

# An Ensemble Random Forest Model for Seismic Energy Forecast

Sukh Sagar Shukla [1], Jaya Dhanya [2], Praveen Kumar [3], Priyanka [3], and Varun Dutt [4]

[1]Research Scholar, school of civil and environmental engineering Indian institute of technology Mandi, Himachal pradesh, India

[2]Assistant professor,school of civil and environmental engineering Indian institute of technology Mandi, Himachal pradesh, India

[3]Research Scholar,school of computing and electrical engineering Indian institute of technology Mandi, Himachal pradesh, India

[4]Professor,school of computing and electrical engineering Indian institute of technology Mandi, Himachal pradesh, India

**Correspondence:** Sukh Sagar Shukla  (D22178@students.iitmandi.ac.in)

**Abstract.** Seismic energy forecasting is critical for hazard preparedness, but current models have limits in accurately predicting seismic energy changes. This paper fills that gap by introducing a new ensemble random forest model designed specifically for seismic energy forecasting. Building on an existing paradigm, provided by Raghukanth et al. (2017), the global energy time series is decomposed into intrinsic mode functions (IMFs) using ensemble empirical mode decomposition for better representation. Following this approach, we split the data into stationary ($IMF_1$) and non-stationary (sum of $IMF_2$-$IMF_6$) components for modeling. We acknowledge the inadequacy of intrinsic mode functions (IMFs) in capturing seismic energy dynamics, notably in anticipating the final values of the time series. To address this restriction, the yearly seismic energy time series is also fed along with the stationary and non-stationary parts as inputs to the developed models. Here, we employed Support Vector Machine (SVM), Random Forest (RF), Instance-Bases learning (IBk), Linear Regression (LR), and Multi-Layer Perceptron (MLP) algorithms for the modelling. Furthermore, the five models discussed above were suitably employed in a novel regression-based ensemble random forest algorithm to arrive at the final predictions. The root mean square error (RMSE) obtained in the training and testing phases of the final model were 0.127 and 0.134, respectively. It was observed that the performance of the developed ensemble model was superior to those existing in literature (Raghukanth et al., 2017). Further, the developed algorithm was employed for the seismic energy prediction in the active Western Himalayan region for a comprehensively compiled catalogue and the mean forecasted seismic energy for year 2024 is $7.21 \times 10^{14} J$. This work is a pilot project that aims to create a forecast model for the release of seismic energy globally and further application at a regional level. The findings of our investigation demonstrate the possibility of the established method in the accurate seismological energy forecast, which can help with appropriate hazard preparedness.

## 1 Introduction

Earthquakes are among the most disastrous natural calamities due to the release of accumulated strain energy from continuous tectonic movements. Like other natural disasters, it can cause destruction both in financial terms and loss of life (Jain, 2016). The devastating potential of earthquakes is increased by their fundamentally unpredictable character due to both aleatory and



epistemic uncertainties (Kramer, 1996; Baker et al., 2021). Whereby, the inherent randomness in the process makes it a real challenge to accurately predict these events. There are several attempts by seismologists to quantify the activity of the regions based on several seismicity indicators. Some of the studies for the Himalayan region are by performing paleo-seismic study (Lavé et al., 2005; Rajendran et al., 2013), statistical inferences (Bilham and Ambraseys, 2005), Global Positioning System (GPS) measurements (Banerjee and Bürgmann, 2002; Ader et al., 2012), numerical (Ismail-Zadeh et al., 2007; Jayalakshmi and Raghukanth, 2017), satellite imagery based data (Bhattacharya et al., 2013; Misra et al., 2020), and, Global Navigation Satellite System (GNSS) studies(Sharma et al., 2023b; Kumar et al., 2023a). However, the inadequacy in precisely monitoring stress changes, pressure, material variability, and temperature variation deep beneath the earth's crust using scientific instruments leads to a lack of comprehensive data regarding accurate seismic characteristics. Subsequently, these lack of information had contributed to the uncertainty in earthquake occurrence, which had resulted to major risk to life and property. Hence, a robust quantification approach is essential considering the increasing vulnerability of the active regions due to developmental activities (Bilham, 2019). However, the variability in seismic behavior, the worldwide occurrence of earthquakes, and the paucity of historical data all hamper predictive modeling. The ethical and practical consequences of delivering earthquake forecasts, the diversity in earthquake magnitudes, and the differences between human and geological timelines all add to the tremendous problem of earthquake prediction (Mignan and Broccardo, 2020; Sun et al., 2022). While progress is being made, the emphasis in earthquake research has turned toward establishing effective earthquake forecast models and early warning systems, understanding seismic risks, and improving preparation to lessen the effects of these deadly occurrences (Bose et al., 2008; Tiampo and Shcherbakov, 2012; Mousavi and Beroza, 2018; Mousavi et al., 2020; Tan et al., 2022). Nevertheless, with the advancements in field instrumentation, once an event occurs, we have attained knowledge to estimate and record its information like magnitude, location, extent of ground shaking, etc., immediately (USGS (2023), IMD (2023)). The robustness of this data has also improved significantly over the years. An intriguing question here shall be, is it possible to predict and be better prepared for a forthcoming event using these information? This work attempts to answer this question by compiling the available earthquake data and implementing it in advanced machine learning (ML) algorithms. ML has evolved so much that its potential is widely explored to address numerous real-world problems (Schmidt et al., 2019; Kaushik et al., 2020; Sarker, 2021; Bertolini et al., 2021; Kumar et al., 2023b). Appropriate data processing using advanced ML algorithms led to successful prediction models. However, ML algorithms have only recently gained popularity in engineering seismology (Xie et al., 2020; Mousavi and Beroza, 2023). The most comprehensive application is in developing efficient Ground Motion Prediction Equations (GMPEs) (Alavi and Gandomi, 2011; Derras et al., 2014; Dhanya and Raghukanth, 2018; Gade et al., 2021; Seo et al., 2022; Sreenath et al., 2024). In another direction, Paolucci et al. (2018) proposed a simple MLP model that should efficiently generate broadband ground motions. Sharma et al. (2023a) improved the model by incorporating source, path and site characteristics. Even though the results from machine learning approaches show promising applications in earthquake engineering, the full potential of ML is yet to be explored in earthquake forecasting. Moreover, the advancements in instrumentation that continuously capture seismic data alongside efficient learning algorithms are expected to reduce the prediction variability considerably. Concerning earthquake prediction, Adeli and Panakkat (2009) employed a probabilistic neural network (PNN) to predict maximum magnitude using eight predefined seismic indicators and found it efficient in predicting





low-magnitude events. Further, Narayanakumar and Raja (2016) divided the events into 15 classes and attempted to predict them employing past 128-year data for the Himalayan Belt using a Back Propagation (BP) algorithm. However, the model seemed inefficient for large events. Furthermore, a few attempts were made with the multilayer perceptron (MLP) model to forecast yearly regional Kavitha and Raghukanth (2016) and global seismic energy releases(Raghukanth et al., 2017). Later, Asim et al. (2018) performed a more extensive study incorporating almost sixty seismic features suitably in the Support vector regression-Hybrid neural network (SVR-HNN) to develop a classification-based prediction model for the occurrence of an earthquake greater than 5. Furthermore, Yousefzadeh et al. (2021) attempted to perform spatiotemporal earthquake magnitude prediction (a classification problem) for Iran using a deep neural network and reported promising model performance. Their analysis was based on shallow neural network (SNN), support vector machine SVM), decision tree (DT), and deep neural network (DNN) models. Additionally, Salam et al. (2021) analyzed earthquake magnitudes for the southern California region using hybrid ANN models and found flower pollination algorithm-extreme learning machine (FPA-ELM) and FPA-SVM gave better predictions. Furthermore, Ridzwan and Yusoff (2023) has also discussed the relevance and evolution of machine learning applications in earthquake predictions.Zhang et al. (2023) presented EPT, a totally data-driven deep learning model. The model use gated feature extraction blocks (GFEB) to extract possible crustal motion and plate movement patterns from worldwide historical seismic data. It utilises them to help anticipate mainshocks in each local and provincial region. Also, Bhatia et al. (2023) proposed a cloud-based edge computing collaborative Internet of Things (IoT) monitoring and prediction system for earthquake prediction. Real-time sensor data was collected using Internet of Things technology and sent to the edge layer, where a unique Bayesian belief model approach was applied to feature categorization. Moreover, the cloud layer earthquake magnitude was predicted using the Adaptive Neuro-Fuzzy Inference System (ANFIS) mechanism. It is also observed that the seismic energy release pattern is one of the prominent indicators in the forecast or prediction models. Hence, considering the uncertainties in the seismic moment release, energy-based models are expected to provide reliable forecasts. Additionally, one can also note that most of the forecast models are based on a specific architecture or hybrid models combining optimization and ML formulations. However, there have been more advancements in the field of ML, and one of the promising addition in the forecast category are the ensemble models (Gastinger et al., 2021). Hence, this work attempts to develop robust forecast model for seismic energy release. The nonstationarity in the time series data is addressed by performing ensemble empirical model decomposition as suggested in Raghukanth et al. (2017). The proposed approach shall be first verified with a global database. Upon suitable validation, the approach shall be extended to developing a forecast model for the active Western Himalayan province. The proposed work is the first of its kind attempt to predict earthquake occurrence in the Himalayan region. The study results are critical in identifying the potential seismic hazard and formulating swift policies for better preparedness for the impending earthquake. A detailed background and formulation of the proposed approach are discussed in subsequent sections.





## 2 Background

The expansion of the global seismograph network in recent decades has sparked a significant increase in the examination of seismic activity. Recognising that defining earthquake size in terms of seismic energy released has greater physical relevance than magnitude alone in understanding the cumulative seismic activity. Hence, the scientific community has worked diligently to predict seismic energy releases on a regular basis. Tsapanos and Liritzis (1992) investigated the relationship between seismic energy release for three seismic regions: Chile, Kamchatka, and Mexico. Tsapanos (1998) used released strain energy to assess

seismic hazards in eleven regions around the world. Moreover, when seismic activities occur in the region where there is potential for other hazards, such as landslides and volcanic eruptions, it leads to multi-hazard scenarios. In this regard, Yokoyama (1988) used the seismic energy release from the precursory earthquake to predict the time of the eruption of dacitic or andesitic volcanoes. Nakamichi et al. (2019) studied the pattern of the rate of seismic energy release for Kelud volcano, Indonesia, before the 2007 effusive dome-forming and 2014 Plinian eruptions. Jaumé and Sykes (1999) have reviewed how seismic energy

release accelerates prior to a great earthquake. They observed that a growing number of cases have been reported in which the occurrence of a large or great earthquake is preceded by an increase in the frequency of moderate-sized earthquakes in the surrounding region. A power-law time-to-failure relationship can be used to model the rate of moment and/or energy release in these sequences. Varga et al. (2012) examined a declustered catalogue of major earthquakes (M >= 7.0) that occurred between 1960 and 2011 and discovered that the latitude distribution of released seismic energy is bimodal with respect to the

equator. Later, Kavitha and Raghukanth (2016) and Raghukanth et al. (2017) have attempted to forecast the seismic energy for the regional and globe scales, respectively, using the artificial neural network technique on mode decomposed energy series. Researchers have also explored the application of seismic energy to determine the probability of the occurrence of earthquake events. Whereby, Zarola and Sil (2018) have predicted the magnitude and time of occurrence of earthquakes for northeast India using four distribution models (Lognormal, Weibull, Gamma, and Log-logistic). Later, Asim et al. (2018) predicted the target

earthquake by employing various seismic features in different machine-learning techniques like support vector regression and hybrid neural networks. Furthermore, Shimony et al. (2020) have studied seismic energy release along the Dead Sea transform (DST) from Intra-Basin Source and its influence on regional ground motions.Recently, Spassiani and Marzocchi (2021) has also described that the energy-based approach is more appropriate in defining the magnitude frequency relation of seismic events due to its physical relevance embedded in rebound theories. In essence, seismic energy is one of the most important

markers of earthquake occurrence, and it follows a pattern in which locations that have been seismically quiet for a long period are more likely to have significant earthquakes. So we can readily say that the previous release of seismic energy can be efficiently used to anticipate the future energy release. As a result, a reliable seismic energy forecasting model can help you prepare for an upcoming occurrence.

Several researchers in the past have extensively explored and utilized various machine learning techniques in earthquake

engineering and related fields. Xie et al. (2020) highlighted the adoption of Multilayer Perceptrons (MLPs) in earthquake engineering. . Raghukanth et al. (2017) utilized a similar model for suitably combining stationary and non-stationary parts of energy series to forecast seismic energy. This technique is also widely used in developing ground motion prediction equations,





as evidenced by the works of Derras et al. (2014); Dhanya and Raghukanth (2018, 2020); Douglas (2021).

Linear Regression (LR) has also been applied in various seismological studies due to its simplicity and efficiency.Pairojn and
Wasinrat (2015) used LR for ground motion prediction in Thailand, while Cho et al. (2022) compared Artificial Neural Net-
works (ANN) and LR for predicting earthquake-induced slope displacement. The Random Forest (RF) technique has similarly
motivated researchers across different fields, including seismology. Asim et al. (2017) used RF to predict earthquake magni-
tude in the Hindukhush region. More recently, Agarwal et al. (2023) developed a hybrid algorithm by combining ANN and
RF to predict the earthquake magnitude.Pyakurel et al. (2023) utilized five supervised algorithms, including RF, to predict
earthquake-induced landslides for the 2015 Gorkha earthquake. Additionally, (Li and Goda, 2023) extended the application of
RF to tsunami early warning systems and loss forecasting. Furthermore, Support Vector Machines (SVM) with the optimized
version named as Sequential Minimal Optimization for regression (SMOreg), as proposed by Shevade et al. (2000), are widely
used for parameter learning. This approach has been applied to various natural hazard contexts, such as flood susceptibility
mapping Saha et al. (2021), ground motion prediction equations (Altay et al., 2023), and landslide monitoring (Kumar et al.,
2023b) .Similar to SMOreg, Instance based learning is also well-explored in earthquake prediction problems, as its reliability
and accuracy owing from algorithm's resistance to noise and outliers, as well as its versatility in the use of distance measures.
Its applicability in seismic prediction is well demonstrated by Reyes et al. (2013), Ghaedi and Ibrahim (2017),Al Banna et al.
(2020), and Ridzwan and Yusoff (2023).

Ensemble models have found versatile applications across numerous domains, including medicine, materials science, and en-
vironmental science. These models, such as those employed by (Tan et al., 2022) for cancer classification using boosted and
bagged decision trees, and by (Rezaei et al., 2022) for predicting gastric cancer using Gradient Boosting Decision Trees (GB-
DTs), Random Forest (RF), Linear Regression (LR), Elastic Net, and LASSO, often outperform individual models. In heart dis-
ease prediction Pouriyeh et al. (2017) (Pouriyeh et al., 2017) and COVID-19 case forecasting Maaliw III et al. (2021)(Maaliw
III et al., 2021), ensemble techniques have also shown superior performance. Their popularity extends to various fields, in-
cluding earthquake prediction in seismology, as seen in (Shishegaran et al., 2019; Joshi et al., 2022). Despite their success,
modeling the forecast of annual seismic energy release remains underexplored, presenting a potential research area. Ensemble
models combine different ML approaches to capture more data variance, thereby outperforming individual models in reliability,
precision, and comprehensibility (Dietterich, 2000; De Gooijer and Hyndman, 2006; Alpaydin, 2007). These models integrate
several base learners and produce diverse ones through generative and non-generative methods(Re and Valentini, 2012).

In light of this, the present study attempts to forecast the annual seismic energy using the advanced ensemble model technique.
In the proposed modelling approach, five non-parametric supervised machine-learning approaches, including Artificial neural
network (ANN), Linear regression (LR), Random forest (RF), Sequential Minimal Optimization regression (SMOreg), and
Instance-Bases learning with parameter k (IBk), are used to construct an individual model to forecast annual seismic energy
separately. In this study, we employ non-generative ensemble methods, combining predictions from multiple individual mod-
els for enhanced accuracy. These foretasted values from each approach are stacked and fed into the ensembled random forest
model to get a more robust and capable model. This method leverages the synergy of findings from many machine-learning
algorithms, demonstrating proficiency in seismic energy predictions. Through the incorporation of pre-learned values as well





as expert views from other approaches, the ensembled random forest model greatly improves its predicting performance. The validity/performance of the approach shall be tested against the global seismic energy forecast model developed by Raghukanth

et al. (2017). Recognising the substantial interest and confidence in such forecasting models by local authorities, policymakers, and government agencies, the study expands its application to the active Western Himalayan area of the Indian subcontinent. This novel approach has the potential to improve our knowledge of and capacity to anticipate seismic energy releases, as well as to benefit stakeholders and support earthquake-prone areas preparedness initiatives.

## 3  Global Seismic Energy (GSE) time series

A comprehensive earthquake catalog is essential for making reliable earthquake predictions. For the present study, the ISC-GEM catalogue (http://www.isc.ac.uk/) utilized by Raghukanth et al. (2017) is considered for model development, comparison and validation of the proposed approach. We used the same inputs as described in Raghukanth et al. (2017) for our analysis. However, for better clarity on the data, we are explaining in brief the processing involved in arriving at the final time series used in modeling.

The global earthquake catalogue contained data spanning from 1900-2015, having a total of 24375 events. The unification of the event magnitude was ensured by converting all reported event magnitudes to moment magnitudes ($M_w$) using suitable empirical relations. Further, the magnitude of completeness (The magnitude above which all earthquakes are reliably recorded) of the catalogue was estimated as 6.4 (Figure 1 (a)) following the maximum curvature method proposed by Wiemer and Wyss (2000). The distribution of the final complete catalog constituting 4619 events used in the analysis performed in this study is

shown in Figures 1 (b) and 1 (c). The catalogue had four great events having an $M_w \geq 9$ and the largest event was the 1960 $M_w$ 9.6 Valdiya earthquake. Furthermore, to estimate the yearly occurrence time series of seismic events it becomes illogical to sum up the corresponding $M_w$. A suitable alternative in agreement with the physics of earthquake occurrence is to convert the given $M_w$ to the corresponding seismic energy and accumulate it to obtain a yearly seismic energy time series. Hence, first the $M_w$ was converted to seismic moment ($M_0$) following Hanks and Kanamori (1979), then the seismic energy ($SE$) was

calculated using the relation proposed by Choy and Boatwright (1995); as expressed further

$$SE = 1.6 \times 10^{-5} M_0 \quad \text{where, } M_0 = 10^{1.5 \times (M_w + 6)} \tag{1}$$

Using the corresponding expression, the yearly seismic energy time series were estimated and the corresponding data is illustrated in Figure 2a. The great events like 1960 $M_w$9.6 Valdiya, 1964 $M_w$9.2 Alaska, 2004 $M_w$9.0 Sumatra, and 2004

$M_w$9.1 Japan earthquakes can be clearly identified from the distinct peaks in the time history. Furthermore, since it might be difficult to capture the energy appropriately wherever there is a sudden jump, the corresponding seismic energy had been converted to log scale as shown in Figure 2b. Additionally, the distribution of the corresponding data is reported to be non-Gaussian and non-stationary (Raghukanth et al., 2017). To perform the predictions better, the corresponding time history is split into stationary and non-stationary parts following an ensemble empirical mode decomposition, as discussed further. It

should be noted that the seismic energies for each incident accumulated annually rather than on any other predetermined time


**Figure 1.** (a) Magnitude of completeness. (b) Magnitude distribution over years. (c) Global distribution of the events from the global earthquake catalogue considered in Raghukanth et al. (2017) adopted for testing the performance of the proposed approach in this work

frame like monthly or weekly because when attempted to collect seismic energy on a monthly basis, for $M \geq 6.4$, certain months had no events, resulting in zero seismic energy. Using a logarithmic representation of seismic energy time series may result in unrealistic results. So accumulation of seismic energy on annual basis is preferred.





## 3.1  Mode decomposition of GSE

The final seismic energy time series were split into orthogonal modes following the empirical mode decomposition (EMD) technique proposed by Huang et al. (1998). The basic functions termed as intrinsic modes were obtained following an iterative procedure on the data directly without any predefined functional form. Hence, the corresponding methodology is reported to be better adaptive of the features in the data. Furthermore, to avoid the issue of mode mixing in conventional EMD Wu and Huang (2009) proposed ensemble EMD (EEMD) where a finite white noise is added to the data while performing decomposition. The

basic steps in the mode extraction involve: (1) Adding finite white noise to the data, (2) Using cubic spline to construct lower and upper envelops connecting consecutive peaks as the respective sides, (3) At each time step estimating the average of positive and negative envelops and then subtract that from the data from step 1, (4) With the data from step 3 steps 2-3 is repeated till we obtain IMF (the time history having the number of extremas and zero-crossing differs by one and the mean is zero), (5) Once first IMF ($IMF_1$) is extracted then the corresponding value is subtracted from the time history in step 1, further follow

steps 2-4 to extract next IMF (6) Repeated until there are no zero-crossings left in the data. To perform ensemble empirical mode decomposition, steps 1-6 are repeated multiple times by adding different white noise to further the mean of IMFs at each level is identified as the final modes. The observations that served as the foundation for the above strategy are (1) If we take an average of white noise in a time domain, it cancels out in the ensemble mean. Hence, in the final noise-added ensembled signal, when averaged, only the signal survives, not the noise. (2) To drive the ensemble to exhaust all viable solutions, finite,

not infinitesimal, amplitude white noise is required. Finite magnitude noise causes the distinct scale signals to reside in the corresponding IMF, as mandated by the dyadic filter banks and therefore improves the meaning of the final ensemble mean. For the data shown in Figure 2b, we were able to extract six IMFs following the described procedure. It is noted that IMFs are mostly uncorrelated and orthogonal. Also, for the physical interpretation of IMFs, various methods exist in the literature to estimate the periodicity of time series, including the instantaneous frequency method and Fourier-based approaches. The

resulting IMFs in the present study are comparable to sine/cosine waves. Hence, counting the number of extremes in an IMF allows for easy estimation of the time period. Table 1 lists the periods of all six IMFs in log-scaled seismic energy time series. Table 1 also includes an estimate of the percentage variance for all IMFs, which is the ratio of IMF variance to data variance. It can be noted that the $IMF_1$ constitute the maximum variance of the time series, and the $IMF_6$ represents the non-stationary trend in the data.  For the forecast of seismic energy, autoregressive modes can be adopted. However, as just using the actual

data might cause greater complexity, the reasonably well-behaved IMF can also be included as variables in the prediction model. Thus, linear and nonlinear parts are separated and while modelling $IMF_1$ is taken separate and remaining $IMFs$ separately as proposed by Iyengar and Raghukanth (2005) and Raghukanth et al. (2017). The corresponding $IMF_1$ is shown in Figure 2c and the sum of $IMF_2$ to $IMF_6$ in Figure 2d. In the present study, this information was suitably incorporated into more advanced machine learning algorithms to make one step ahead of seismic energy forecast. A detailed description of the

ML algorithms and the corresponding implementation is discussed further

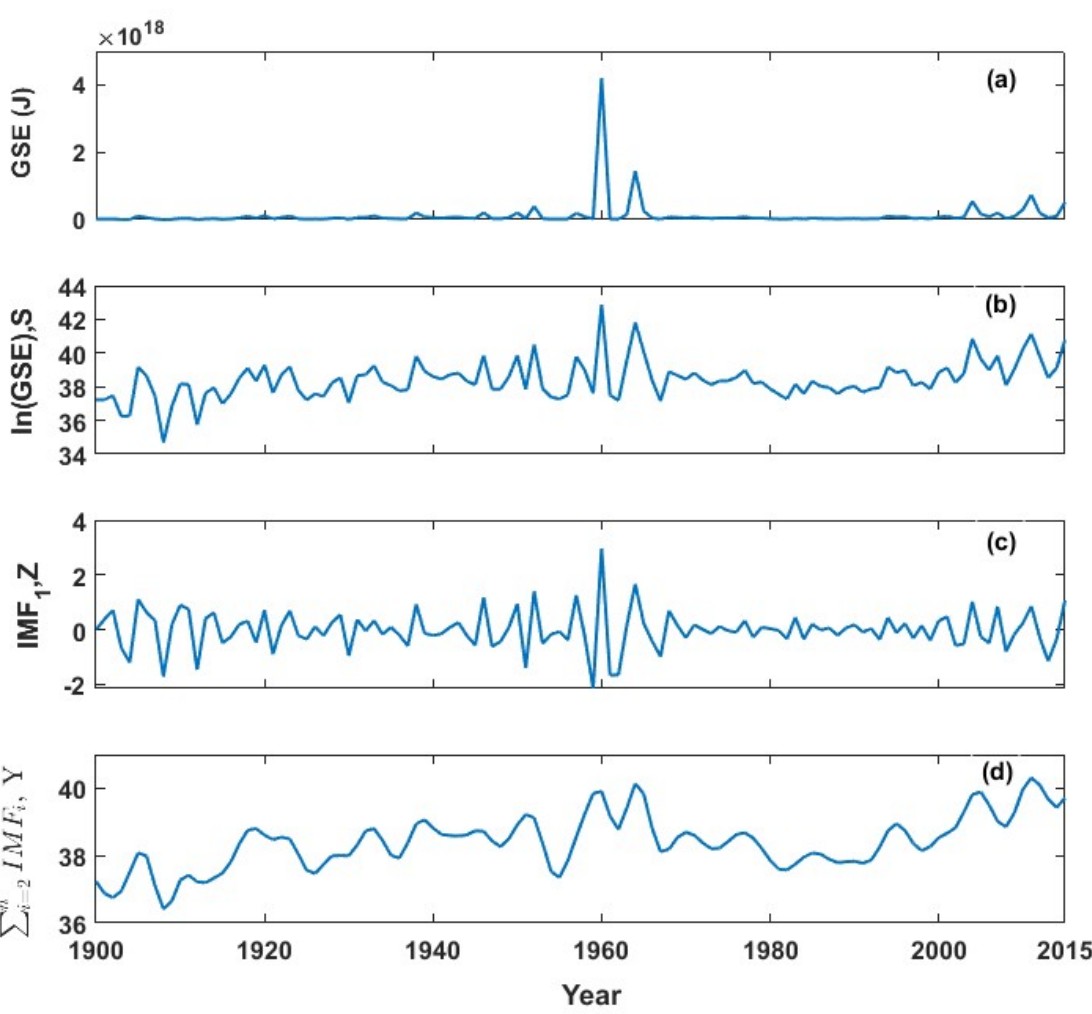

**Figure 2.** (a) Estimated Global seismic energy (J) time series from ISC gem catalogue used in developing the models (b) log scaled Global seismic energy time series (ln(GSE)) (c) First intrinsic mode estimated from ln(GSE) by performing ensemble empirical mode decomposition (EEMD) (d) Sum of second to last intrinsic modes estimated from ln(GSE) by performing EEMD.



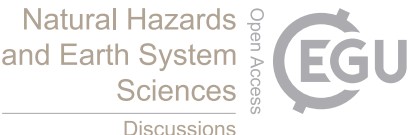

**Table 1.** Period observed and the variance captured by the IMFs obtained for log scaled seismic energy time series

| IMFs | log(Global seismic energy time series) | | Log seismic energy time series for Western Himalaya | |
| --- | --- | --- | --- | --- |
| | Period (Years) | % (Variance) | Period (Years) | % (Variance) |
| $IMF_1$ | 2.95 | 49.18 | 2.76 | 74.16 |
| $IMF_2$ | 6.29 | 7.02 | 8.29 | 16.57 |
| $IMF_3$ | 11.55 | 5.61 | 10.6 | 4.50 |
| $IMF_4$ | 31.00 | 6.43 | 26 | 2.12 |
| $IMF_5$ | 91 | 6.60 | — | 1.97 |
| $IMF_6$ | — | 25.80 | — | — |





## 4   Methodology

There are numerous advanced machine-learning techniques available in the literature. Some of the widely used variants include Artificial neural networks (ANN), Decision trees, Instance-based learnings, classification and regression models Bishop (2016). In this study, we attempted to include each of these flavours by including one representative algorithm for the analysis and

further combining them using a suitable ensemble formulation. The description of the models utilised in the study is provided further.

### 4.1   Multi-Layer Perceptron

Multi-Layer Perceptron (MLP) comes as part of ANN Bishop (2016). A typical MLP architecture constitutes 3 layers, which are input, hidden, and output, respectively, mutually interconnected with weights. The typical functional form of an MLP from

a single layer can be represented as

$$\widehat{y} = f(Wx + b) \tag{2}$$

where $\widehat{y}$ is the output from the layer, $f$ the activation function, $W$ the weights, $x$ the vector of inputs corresponding to the values at the previous layer, and b the bias. The number of hidden layers, nodes, and activation functions (e.g., linear, logistic,

tanh, ReLU) depend on the non-linearity between predicted and predictor variables (Kumar et al., 2023b). Once the architecture is finalized, parameters are estimated using back-propagation with mean squared error (MSE) or mean absolute error (MAE) as the cost function. Our MLP uses four input features: log seismic energy (S), IMF1 (Z), $\sum_{i=1}^{n} IMFi$ (Y), and the year of occurrence of seismic energy. The hyperparameters for the model were optimized by varying the parameters as shown in Table 2. In time-series forecasting, the concept of lag values is fundamental. Lag values refer to the number of past observations used

to predict future values in a time series (Surakhi et al., 2021). By incorporating information from previous time points, models can capture temporal dependencies and trends, leading to more accurate forecasts. The lag values were varied from 1 to 15, the number of neurons in hidden layers from 1 to 15, the learning rate (L) from 0.1 to 1.0, momentum (M) from 0.1 to 1.0, and the number of epochs from 100 to 2000. The batch size was consistently set to 100.

### 4.2   Linear Regression

LR is one of the widely used statistical machine learning model Brownlee (2023). The model establishes a linear relation between the target variable and the input features. The general form can be expressed as

$$\widehat{y} = \beta_o + \beta_1 x + .... + \beta_n x_n \epsilon \tag{3}$$

$$MSE = \sum_{i=1}^{p} (y - \widehat{y})^2 \tag{4}$$

where y is the target variable and $\widehat{y}$ is the predicted variable by LR, $x_1, \ldots, x_n$ are the input variables, $\beta_0$ is the intersect, $\beta_1$

$, \ldots, \beta_n$ are the slope or LR coefficient, n is the number of features, p is the total number of datapoints, and $\epsilon$ the error. Further,





based on the number of input variables, there are two variants of LR, which are single-LR and multiple-LR. The unknowns $\beta_0$ ,...,$\beta_n$ in the model are estimated by gradient descent algorithm with MSE as the cost function.

Linear Regression (LR) architectures are employed for developing the forecast model. The inputs are the same as for the MLP. The lag values were varied from 1 to 15, which resulted in transformed input parameters from the higher order of the time

variable and the product of time and different lagged variables. Attribute selection methods, such as the M5 method and the Greedy method can be used to reduce the number of attributes. In this work, the M5 method was used, retaining only the time steps from input variables that significantly affect the results for regression. The hyperparameter Ridge was varied from $1.0 \times 10^{-6}$ to $1.0 \times 10^{-9}$, and the batch size was consistently set to 100, as shown in Table 2.

### 4.3 Random Forest

Random Forest (RF) is a supervised learning model used for classification and regression (Breiman, 2001a). It combines multiple decision trees to improve predictive accuracy (Cutler et al., 2012). A decision tree has decision nodes (with branches) and leaf nodes (with no branches). Trees start from a root node containing the entire dataset, splitting at each node based on attribute selection measures (ASM) like information gain or Gini index. Pruning removes unnecessary nodes to prevent

overfitting.

RF addresses overfitting by building multiple decision trees on different data subsets and averaging their predictions. This ensemble of uncorrelated trees uses bootstrap sampling and feature randomness. RFs have lower computational costs, handle missing data, and can manage larger datasets efficiently. Randomness in tree generation is controlled by a fixed seed (set to 1). The number of trees (iterations) was varied from 30 to 120, and tree depth is unlimited (depth of 0). Features at each split are

calculated by $(\log_2(\text{N}) + 1)$, where N is the number of predictors (Breiman, 2001b). The lag values were varied from 1 to 15, and the batch size and bag size were consistently set to 100, as shown in Table 2.

### 4.4 Sequential Minimal Optimization regression

Sequential Minimal Optimization (SMO) is an iterative algorithm proposed by Platt (1998) for solving regression problems using support vector machines (SVM). SMO simplifies the optimization problem by breaking it down into smaller sub-problems

that can be solved analytically, which makes it more efficient for training SVMs. Further improvements to SMO for regression were proposed by Shevade et al. (1999), who introduced modifications to enhance its efficiency. These improvements address the way SMO updates and maintains threshold values, resulting in two significantly more efficient versions for regression tasks. The corresponding algorithm effectively solves the quadratic optimization problem inherent in SVM training. SMOreg employs four input features: log seismic energy (S), IMF1 (Z), $\sum_{i=1}^{n} IMFi$ (Y), and the year of occurrence of seismic energy.

The model is optimized by fine-tuning hyperparameters such as the complexity number ($C$) and epsilon ($\epsilon$). In this study, the complexity number was varied from 1 to 9 to balance minimizing training error and problem complexity. Epsilon was varied from $1.0 \times 10^{-9}$ to $1.0 \times 10^{-15}$ to determine the allowable error within the epsilon tube. The kernel function, crucial for SVMs, impacts the ability to manage complex relationships in the data. Various kernel functions, including polynomial (Polykernel),




puk, RBF, and string, were considered. The lag values were varied from 1 to 15, and the batch size was consistently set to 100,
as shown in Table 2.

## 4.5   Instance-Bases learning with parameter k

Instance-based learning (IBL), also called instance-based learning with parameter k (IBk), is a type of unsupervised learning
used for both classification and regression problems Aha et al. (1991); Jo and Jo (2021). It falls under lazy learning algorithms,
which memorize the training data and make predictions based on the similarity between new and training datasets. The parameter k represents the number of nearest neighbors considered for predictions. IBk searches for the k most similar instances from
the training dataset based on the similarity measures using Manhattan distance, Euclidean distance or other distance matrices.
More accurately, let given instance x be described by the feature vector. The euclidean distance between $x_i$ and $x_j$ is given by

$$d(x_i, x_j) \equiv \sqrt{\sum_{r=1}^{n} ((a_r(x_i) - a_r(x_j))^2)} \tag{5}$$

Here in the K-Nearest Neighbour algorithm, the target function may be either real-valued or discrete- valued defined by $\widehat{f}(x_q)$,
which is just the most common value of $f$ among k training example nearest to $x_q$.

$$\widehat{f} \leftarrow \frac{\sum_{i=1}^{k} f(x_i)}{k}) \tag{6}$$

In this study, the k value was varied between 1 and 12, using Euclidean and Manhattan distances. The lag values were varied
from 1 to 15, resulting in transformed input features. The number of nearest neighbors for prediction was set using these variations, with a consistent batch size of 100, as shown in Table 2. The LinearNNSearch algorithm, suitable for small datasets,
was adopted for nearest neighbor search, employing a linear search across all data points.

## 5   Ensemble Models

Hyndman and Athanasopoulos (2018) suggested that in time series forecasting approaches there is a need to include relevant
characteristics to increase accuracy. Also, the wider notion of adding time-related characteristics is a well-known approach in
machine learning and forecasting. Hence, the inclusion of year as one of the input features is decided in the present study. Now
using only IMFs as the inputs for developing forecasting model might not be that effective because Finding the IMF 1 at the
conclusion of data can be challenging because to the lack of defined envelopes on both sides. To address this challenge, the
previous value of the end point might be used as the next value. However, this strategy is not effective for anticipating issues. If
n data is provided, IMFs can only be extracted for i = 2, 3, 4,..., n - 1. As the distance between extrema increases, extrapolation
errors might permeate into the signal, misrepresenting greater IMFs at the end points. Hence The inclusion of S in the input
ensures that the deficiency of the empirical mode decomposition to estimate the end value does not affect the model predictions.
Hence, The time histories from S, Y , and Z (see Figure 2) were provided simultaneously as inputs to the model along with




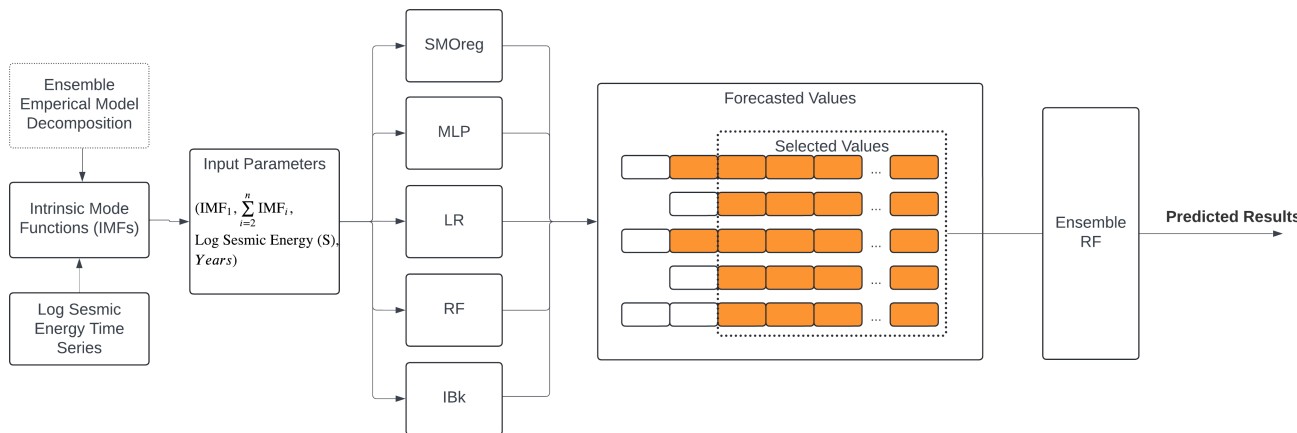

**Figure 3.** The flow chart representing the various steps and modelling approaches adopted for seismic energy forecast

the year to predict S as the output variable. The time series from up to 1995 is taken for the training phase and the remaining data from 1996-2015 is considered for the testing phase. Additionally, the combination of the inputs is determined such that

it gives the best prediction performing beta coefficient analysis. Whereby the approach tries different indices and retains only those that are significant and decreases the error. To determine the optimal lag period, an analysis was performed by varying the lag from 1 to 15 time points. The optimal lag value was identified by evaluating the model's performance and selecting the lag period that minimizes the prediction error. The range of lag values, along with the other hyperparameters varied to get the final model, are presented in Table 2. Also, different lag values for different models were obtained, and these values, along

with the other hyperparameters used in the prediction models, are present in Table 3. The overall flow of the proposed approach is described through Figure 3. Furthermore, a detailed description of the model architectures of ensemble model is furnished further under section 5.1.



## 5.1 Ensembled model architecture

Ensembling can be performed by second-level trainable combiners through meta-learning techniques Duin and Tax (2000) In
the present study, the stacking method was employed, wherein the output results of the base or weak learners were used as
features in an intermediate space. These features were subsequently fed as input to a second-level meta-learner to perform
a trained combination of weak learners. The base learners in this study included MLP, LR, RF, SMOReg, and IBk, which
forecasted the values of seismic energy, as depicted in the figure 4. These base learners were optimized models with varying
parameters, as detailed in Table 2. Each base learner produced predictions of different lengths due to the use of varying lag
values in their optimization process. To address this discrepancy and ensure consistency across all learners, we considered the
shortest prediction length among the base learners. This approach was visually represented in Figure 3, where the orange blocks
indicated the forecasted values and the empty blocks denoted absent values. Here, forecasted values from all five techniques
were then used as input features for the ensemble RF model, with the actual log seismic energy as the target for regression.
This ensemble RF model ultimately predicted the log seismic energy, integrating the results from the base learners to enhance
prediction accuracy.

To ensure optimal performance of the models, we employed grid search techniques for hyperparameter tuning. Grid search is
an exhaustive search method that tests all possible combinations of specified hyperparameters to identify the best-performing
configuration for each model. The process involves defining a parameter grid for each model, specifying a range of values for
each hyperparameter. For example, for the Multi-Layer Perceptron (MLP), the grid includes different numbers of neurons in
the hidden layers, learning rates, and momentum values. For the Random Forest (RF), the grid includes the number of trees
and maximum depth. Each combination of hyperparameters is evaluated using an 80:20 train-test split method. The dataset
is divided into 80 % training and 20 % testing sets, with the model trained on the training set and evaluated on the testing
set. The performance of each hyperparameter combination is assessed using an appropriate evaluation metric, such as mean
squared error (MSE) or root mean square error (RMSE). The combination of hyperparameters that results in the lowest error
(or highest accuracy) on the training set is selected as the optimal configuration for the model. The final model, with optimized
hyperparameters, is then tested on the testing dataset to evaluate its generalization performance. This ensures that the selected
model configuration not only performs well on the training data but also maintains its accuracy and robustness on unseen data.





**Table 2.** Combinations considered for the optimization of hyper-parameters in the model architecture

| Model | Parameter | Range of parameter | |
|---|---|---|---|
| | | Global | Western Himalaya |
| MLP | Lag | 1 to 15 | 1 to 12 |
| | Hidden layers | 1,2 | 1,2 |
| | Neuron in hidden layers | 1 to 15 | 1 to 15 |
| | Learning Rate | 0.1 to 1.0 | 0.1 to 1.0 |
| | Momentum | 0.1 to 1.0 | 0.1 to 1.0 |
| | Batch size | 100 | 100 |
| | Epochs | 100 to 2000 | 100 to 2000 |
| Linear Regression | Lag | 1 to 15 | 1 to 12 |
| | Ridge | 1.0E-6 to 1.0E-9 | 1.0E-6 to 1.0E-9 |
| | Batch size | 100 | 100 |
| Random Forest | Lag | 1 to 15 | 1 to 12 |
| | Batch size | 100 | 100 |
| | Bag size | 100 | 100 |
| | Number of trees | 30 to 120 | 30 to 120 |
| SMOreg | Lag | 1 to 15 | 1 to 12 |
| | Kernel | Poly, puk, RBF, string | Poly, puk, RBF, string |
| | Epsilon (E) | 1.0E-9 to 1.0E-15 | 1.0E-9 to 1.0E-15 |
| | Complexity | 1 to 9 | 1 to 9 |
| | Batch size | 100 | 100 |
| Ibk | Lag | 1 to 15 | 1 to 12 |
| | K | 1 to 12 | 1 to 12 |
| | Distance Function | Euclidean and Manhattan | Euclidean and Manhattan |
| | Batch size | 100 | 100 |





**Table 3.** Optimized hyper-parameter of the models for the data under consideration

| Model | Parameters | Global | Western Himalaya |
|---|---|---|---|
| MLP | Lag | 8 | 6 |
| | Hidden layers | 2 | 1 |
| | Neuron in hidden layer | 4,2 | 11 |
| | Learning Rate | 0.3 | 0.3 |
| | Momentum | 0.2 | 0.2 |
| | Batch Size | 100 | 100 |
| | Epochs | 1500 | 500 |
| Linear Regression | Lag | 6 | 6 |
| | Ridge | 1.0E-8 | 1.0E-8 |
| | Batch Size | 100 | 100 |
| Random Forest | Lag | 7 | 7 |
| | Batch Size | 100 | 100 |
| | Bag size | 100 | 100 |
| | Number of Trees | 100 | 100 |
| SMOreg | Lag | 8 | 5 |
| | Kernel | Poly | Poly |
| | Epsilon (E) | 1.0E-12 | 1.0E-12 |
| | Complexity | 1 | 1 |
| | Batch Size | 100 | 100 |
| Ibk | Lag | 8 | 8 |
| | K | 2 | 9 |
| | Distance Function | Euclidean | Euclidean |
| | Batch Size | 100 | 100 |




## 6   Validation

Based on the optimized hyper-parameters, the predictions from the models in the training and testing phases are summarized in Figure 4 for the individual models. The performance is observed to vary between models both in training and testing parts. Hence, to have a quantitative evaluation of the model performances, the following indicators are estimated for both the training and testing phases:

1. Standard deviation of error ($\sigma(\epsilon)$)

$$\sigma(\epsilon) = \sqrt{\frac{\sum\limits_{i=1}^{N}(S_i - \hat{S}_i) - \overline{(S_i - \hat{S}_i)}}{N-1}} \quad (7)$$

2. Pearson correlation coefficient ($R$)

$$R = \frac{\sum\limits_{i=1}^{N}(S_i - \overline{S_i})(\hat{S}_i - \overline{\hat{S}_i})}{\sqrt{\sum\limits_{i=1}^{N}(\hat{S}_i - \overline{\hat{S}_i})^2 \sum\limits_{i=1}^{N}(S_i - \overline{S_i})^2}} \quad (8)$$

3. Performance Parameter ($PP$)

$$PP = 1 - \frac{\langle \| S - \hat{S} \|^2 \rangle}{\sigma_{\hat{S}}^2} \quad (9)$$

4. Root Mean Squared Error ($RMSE$)

$$RMSE = \sqrt{\frac{\sum\limits_{i=1}^{N}(S_i - \hat{S}_i)^2}{N}} \quad (10)$$

The corresponding estimations for the weak-learners and ensemble models are summarised Table 4. Out of the models, the MLP model performs well in the training phase; however, at the testing phase, it is relatively weak. The RF model also shows a similar trend to that of MLP. However, LR and SMOreg models are observed to perform consistently in both the training and testing phases. However, IBk architecture is the least-performing model for the data under consideration. Nevertheless,

according to a detailed literature review explained earlier, ensemble models are expected to improve the model's performance. Thus, a suitable ensemble model is developed as described in section 5. The corresponding model performance is summarized in Figure 5 and Table 4.

It is interesting to note that the resultant ensemble model outperformed the weak learners. Furthermore, the model performance is good and consistent in both the training and testing phases. Additionally, from a comparison of performance with

the previous study (Raghukanth et al., 2017) (Table 4) on the same data, we were able to conclude that the ensemble model performs significantly better, having lesser variability. Thus, the corresponding model can be suitably implemented in making



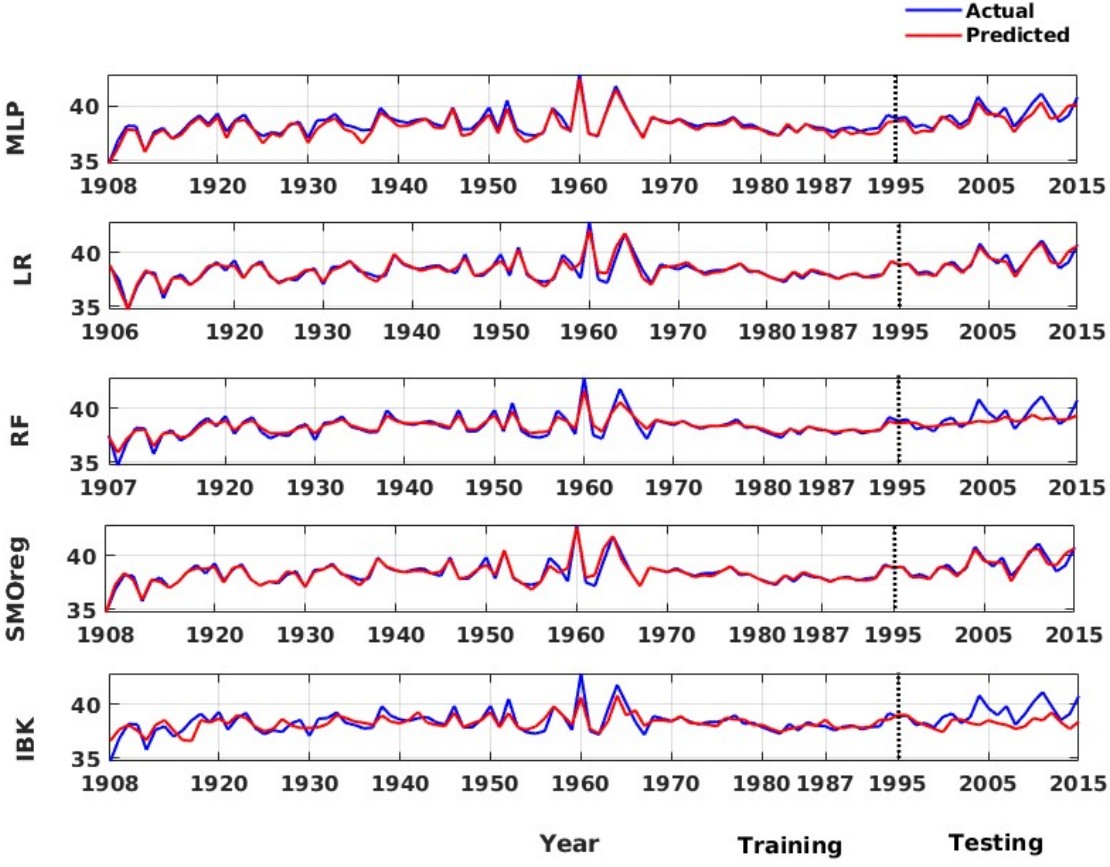

**Figure 4.** Actual vs Predicted values of global log seismic energy from individual machine learning technique adopted in this work

reliable seismic energy predictions and thus has significant application in earthquake forecasts. With this motivation, we further attempted to explore the proposed approach for regional-level seismic energy forecast. The active Western Himalayan province is chosen for the evaluations. A detailed description of the corresponding data, processing and modelling is provided under

section 7.

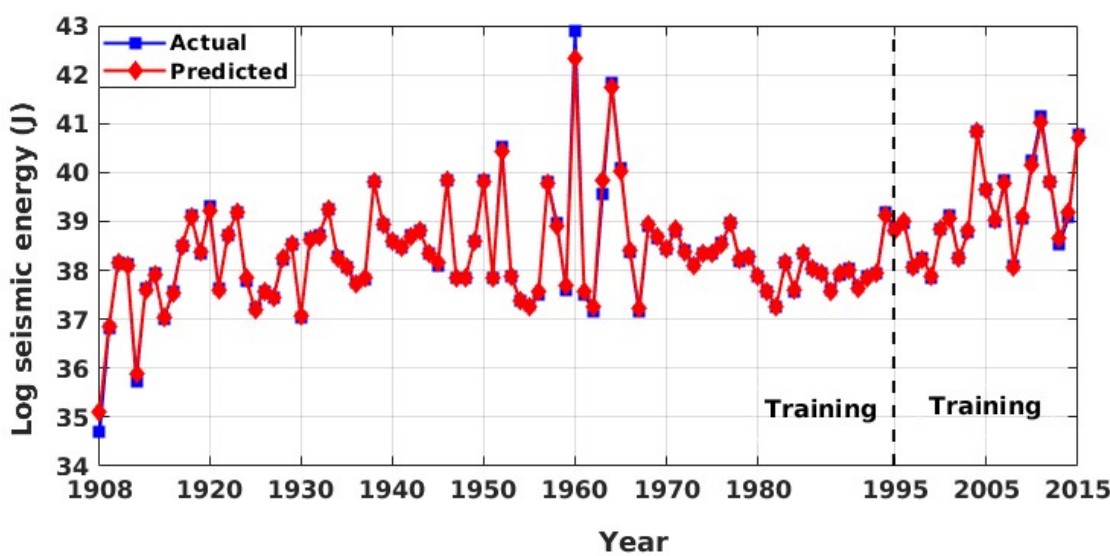

**Figure 5.** Actual vs Predicted values for global seismic energy from proposed ensembled random forest model

**Table 4.** The performance evaluation of individual models and the final ensemble random forest model at the training and testing phases of Global seismic energy

| Models | Training | | | | Testing | | | |
|---|---|---|---|---|---|---|---|---|
| | $\sigma(\epsilon)$ | $R$ | $PP$ | $RMSE$ | $\sigma(\epsilon)$ | $R$ | $PP$ | $RMSE$ |
| Raghukanth et al. (2017) | 0.285 | 0.968 | 0.920 | 0.284 | 0.361 | 0.940 | 0.860 | 0.364 |
| Present Study | | | | | | | | |
| MLP* | 0.259 | 0.971 | 0.887 | 0.362 | 0.497 | 0.862 | 0.611 | 0.607 |
| Linear Regression (LR) | 0.347 | 0.946 | 0.896 | 0.345 | 0.373 | 0.926 | 0.855 | 0.371 |
| Random Forest (RF) | 0.378 | 0.974 | 0.877 | 0.377 | 0.789 | 0.693 | 0.205 | 0.868 |
| SMOreg** | 0.309 | 0.958 | 0.919 | 0.307 | 0.400 | 0.916 | 0.837 | 0.393 |
| IBk*** | 0.642 | 0.819 | 0.649 | 0.639 | 0.931 | 0.323 | -0.808 | 1.308 |
| **Ensemble RF** | **0.127** | **0.994** | **0.986** | **0.127** | **0.136** | **0.992** | **0.981** | **0.134** |

\* MLP- Multi-Layer Perceptron; \*\*SMOreg - Sequential Minimal Optimization regression;

\*\*\*IBk - Instance-Bases learning with parameter k





## 7 Application to Western Himalayas

The vast Indian subcontinent region, which includes Bangladesh, India, Nepal, Bhutan, Pakistan, and Sri Lanka, is prone to frequent and severe earthquakes. This region is particularly vulnerable to seismic activity because of the tectonic moments and the proximity to the convergent margin of the Indian and Eurasian plates. Whereby the subsequent collision has resulted in a vast mountain belt known as the Great Himalayas, where frequent earthquakes are caused by ongoing tectonic activity. The uplift due to collision has caused linear zones of deformation, leading to crustal shortening along major boundary faults. These faults are Himalayan Frontal Thrust (HFT), Main Boundary Thrust (MBT) and Main Central Thrust (MCT), which resulted in some large paleo-earthquakes in the region (Geological Survey of India (GSI), 2000). India's northern and northeastern parts are more vulnerable; they are classified majorly as seismic zone IV and V in IS:1893-1 (2016), which indicates the highest degree of seismic hazard. As the Indian plate slowly sinks and subducts beneath Asia at a pace of around 47 mm/year, this collision tectonics makes the area very vulnerable to catastrophic earthquakes due to energy accumulation and subsequent release Bendick and Bilham (2001). Several large earthquakes has been observed in this region in last two decades, such as the Uttarkashi earthquake in 1991 ($m_b$ 6.6), the Chamoli earthquake in 1999 ($m_b$ 6.3), the Kashmir earthquake in 2005 ($M_w$ 7.8), the Sikkim earthquake in 2011 ($M_w$ 6.9), Nepal earthquake in 2015 ($M_w$ 7.8). Moreover, the region between Kangra earthquake in 1905 ($M_w$ 7.8) and Bihar-Nepal earthquake in 1934 ($M_w$ 8) is relatively silent and hence is identified as the central seismic gap (CSG) region having potential of generating $> 8$ magnitude event (Bilham et al., 1997; Khattri, 1999; Bilham, 2019). It is, therefore, essential to have a reliable quantification of hazard in order to reduce the related seismic risks in the active western Himalayan area. The current approach, designed especially for this seismically active and dynamic area, is critical. Its alignment with the distinct geological features of the western Himalayas emphasises its significance and makes it an indispensable instrument for reducing the possible effect of seismic occurrences in this susceptible region. Furthermore, seismic hazard studies have also reported high values for design parameters for the region due to its active tectonics and recurrence rate (NDMA, 2010; Nath and Thingbaijam, 2012; Dhanya and Raghukanth, 2022; Sreejaya et al., 2022). Considering the tectonics and the risk due to exposure an efficient forecast model is critical for this region. However, such a model is not attempted for the region. Hence, the present work aims to develop a robust forecast model for annual seismic energy release in the Western Himalayas. The ensemble model algorithm validated in Section 6 shall be utilized for the work. A detailed description of the data compiled for the region and the resultant forecast is furnished further.

### 7.1 Study region and data preparation

From the tectonics described earlier, the Hindukhush region and the adjoined region are seismically very active, as evident from Figure 6. This observation is consistent with both the frequency of documented earthquakes and the geographical distribution of faults and lineaments (Figure 6). This has motivated researchers to divide the whole geographical area of the country and the adjoining region into different seismic zones. For instance, Khattri et al. (1984), used seismotectonic and seismicity data to split the nation into 24 source zones. While Bhatia et al. (1999) found 85 source zones in India, 40 seismic zones were detected by another research Parvez et al. (2003). Considering all these past efforts The National Disaster Management Authority (NDMA)


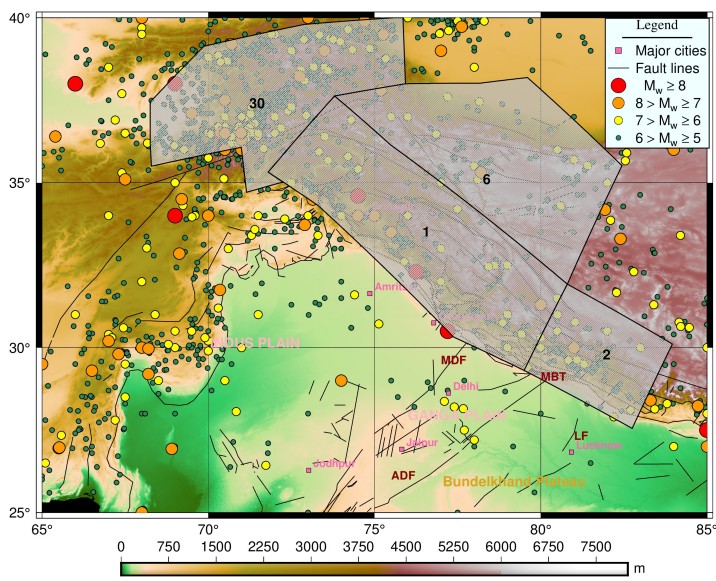

**Figure 6.** The regional level tectonics and the seismogenic zones as per NDMA (2010) in Western Himalayas

of India created a thorough study in 2010 that further divided the whole Indian subcontinent into 32 seismic zones, designated

SZ-1 to SZ-32 NDMA (2010). This division of seismic zones was done by considering factors such as regional geodynamics, fault alignment, and recurrence parameters for the regions. Among these 32 zones, the present study focuses on earthquakes in SZ-1, SZ-2, SZ-6, and SZ-30, which lie in the Western part of the Himalayan belt. As this is preliminary work towards energy prediction for the regions, the comprehensive catalogue is combined together for all zones in the Western Himalayas. Here, the earthquake catalog for the region has been taken from Dhanya et al. (2022) and was updated until December 31, 2023,

via the USGS seismic database (https://earthquake.usgs.gov/earthquakes/search/). There are 25,769 events (for the western Himalayan region considered for this work) in the final updated catalog spanning from 1250 BC to 2023 AD. Furthermore, the updated catalogue has been checked for both completeness for year and magnitude. For completeness of year, the method suggested by Stepp (1972) has been adopted in which the standard deviation of mean rate is plotted as a function of sample length and the period where this value deviated from tangent, i.e., $(1/\sqrt{(T)})$ is consider as completeness for considered

magnitude. Furthermore, the magnitude of completeness was identified from the maximum curvature method proposed by Wiemer and Wyss (2000). Whereby the catalogue compiled for the region in the present work is observed to have a magnitude of completeness of $M_w$ 4 and the corresponding year of completeness as 1964 (Figure 7 (a) and 7 (b)). Hence, events spanning from 1964 and having a magnitude greater than 4 $M_w$ have been considered for further input preparation. The distribution of the event in the final compiled catalogue can be identified from Figure 8.

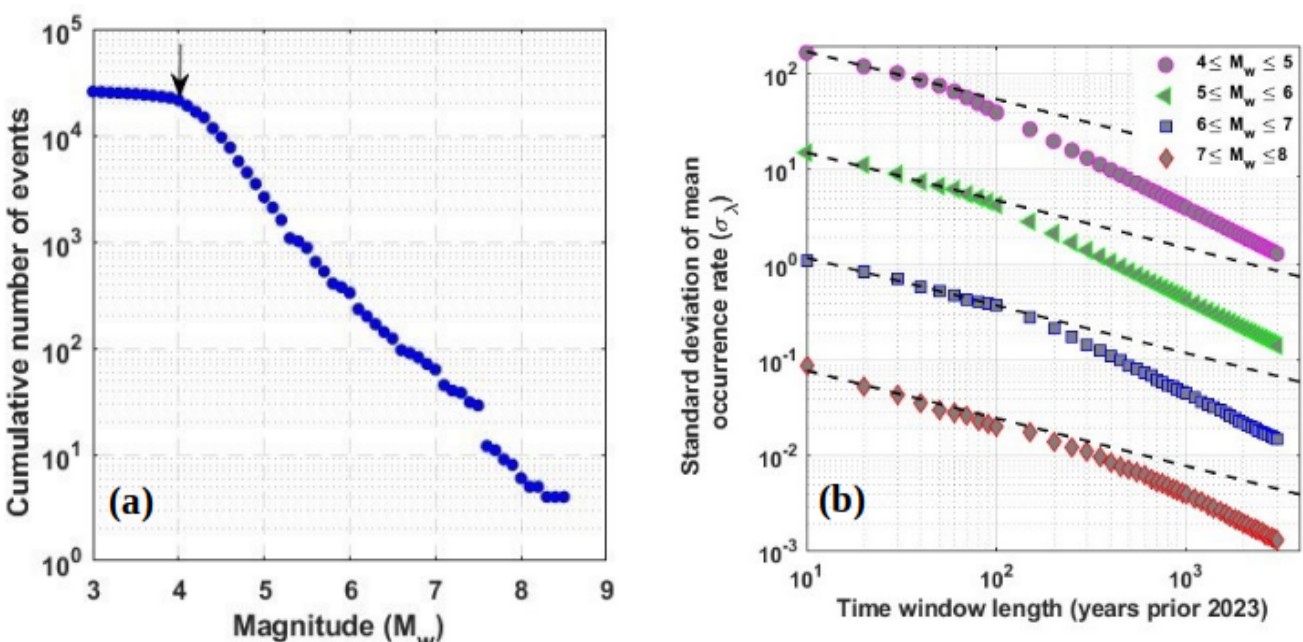

**Figure 7.** (a) The magnitude of completeness (Wiemer and Wyss, 2000) and (b) year of completeness (Stepp, 1972) obtained for the catalogue compiled for Western Himalaya region


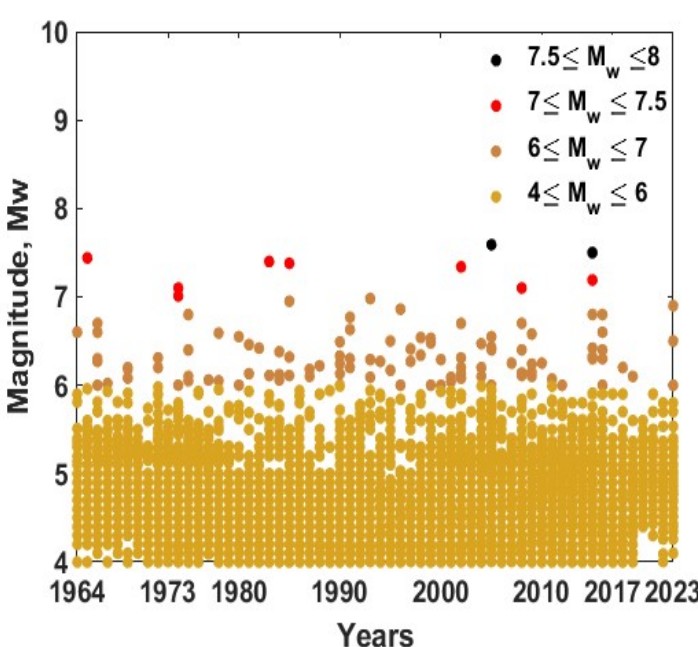

**Figure 8.** Distribution of events for the complete catalog for the western Himalayan region. Events with magnitude equal to or greater than 7.5 $M_w$ are shown with black legends.





## 7.2 Seismic time series and mode decomposition

The same approach discussed under section 3 has been adopted for Western Himalaya, where the complete catalogue spanning from 1964-2023 with 20774 events having a magnitude in different scales is unified by converting all the earthquake magnitude into moment magnitude $M_w$. The catalogue has two major earthquakes having a magnitude $\geq 7.5$ $M_w$, the 2005 Kashmir earthquake (7.6 $M_w$) and the 2015 Afghanistan earthquake (7.5 $M_w$) (Figure 8). After unification, magnitudes are converted to seismic energy using Eqn. 1 discussed under section 3 considering the physical significance of the parameter in earthquake occurrence. Furthermore, energies are added annually to get the seismic energy time series. From Figure 9 (a), one can note two distinct peaks at 2005 and 2015 indicator of the major earthquakes explained earlier. Furthermore, to enhance the predictability of the time series, these values are converted to log scale to remove sudden jumps similar to that described in Section 3. After obtaining the seismic energy time series, it further decomposes into intrinsic mode functions using the EEMD technique. The corresponding division is expected to account for the linear and non-linear components of time histories appropriately. Thus, by applying the EEMD technique as described in section 3.1 on the log-scaled Western Himalaya seismic energy time series (Figure 9 (b), first subplot), we were able to obtain five intrinsic mode functions (Figure 9 (b)). Furthermore, the correlation coefficients between the models are presented in Figure 10. It is noted that IMFs are mostly uncorrelated and orthogonal. Table 1 lists the periods and variance of all five IMFs in log-scaled seismic energy time series. Similar to the global seismic energy modes, for the regional model also, the period of the IMFs seems to increase. Furthermore, the $IMF_1$ is observed to capture maximum variance in the data. To incorporate these IMFs as input for the machine learning techniques, $IMF_1$ and the sum of $IMF_2$ to $IMF_5$ ($\sum_{i=2}^{5} \text{IMF}_i$) have been taken separately as suggested in Raghukanth et al. (2017). Furthermore, considering the limitation of the EEMD approach in predicting the last value appropriately as similar to that discussed in section 6, the log seismic energy and year of occurrence is also considered as inputs into the machine learning models. The corresponding information is illustrated in Figure 11. A detailed description of the model architecture that was found optimal for the regional database is discussed further





**Figure 9.** (a) Annual Seismic energy estimated for the catalogue compiled for western Himalayas (b) Log scaled seismic energy for western Himalayas and the corresponding intrinsic mode function obtained by employing ensemble empirical mode decomposition (EEMD)





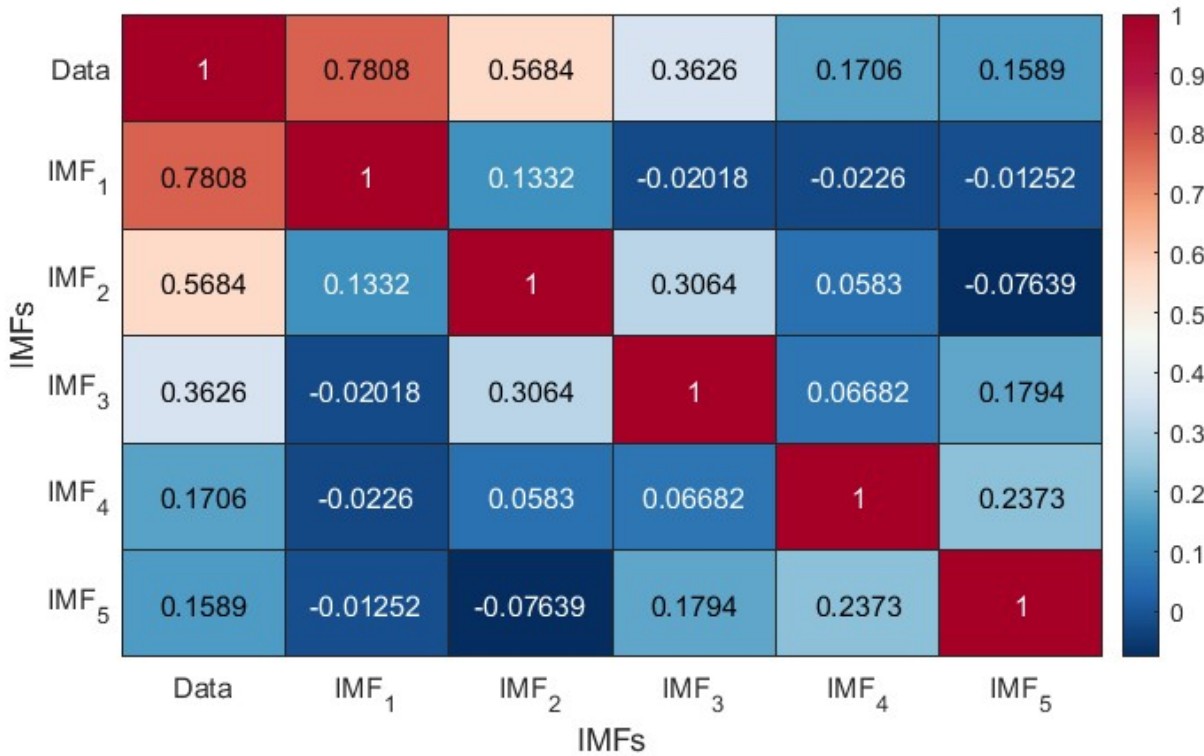

**Figure 10.** The correlation coefficient estimated for the intrinsic mode functions extracted from log scaled seismic energy time series of Western Himalayas



**Figure 11.** Estimated seismic energy series from updated catalogue for Western Himalayan region used in developing the models (a) Global seismic energy (GSE) time series having units as joules (b) log scaled GSE (c) First intrinsic mode estimated from ln(GSE) by performing ensemble empirical mode decomposition (EEMD) (d) Sum of second to last intrinsic modes estimated from ln(GSE) by performing EEMD

### 7.3 Model architecture for Western Himalayas

After input preparation, the approaches discussed under Section 4 are also tested for the active Western Himalayan region, i.e., obtained IMFs along with the log seismic energy and year of occurrence are taken as input for the first-level individual machine learning techniques (MLP, LR, RF, SMOreg, and IBk) and the forecasted results of these techniques as an input to the final





ensemble random forest technique (Figure 3). For lag consideration look back period is varied from 1-15 for the individual models and the value for which results are optimum is selected. Other hyper parameteres were also suitably iterated to find the best model in each individual architecture for the data under consideration. The lag value for individual models, along with the other hyper-parameters used to optimise the model predictions for the regional dataset corresponding to various techniques, are

present in Table 3. For the final ensemble RF model 100 trees were considered along with the depth of the tree as 0 and both bag and batch size also as 100. Furthermore, for the ensemble model, the overall lag value of 8 is adopted. For base learners, the time series data is divided into 80 to 20 % for training and testing, respectively, i.e., time series up to 2011 is used to train the model, and from 2012 to 2023 is used to test the model.

## 7.4   Results for Western Himalaya

The representation of the model results and the comparison with the data is illustrated in Figure 12 for the individual model and Figure 13 for the ensemble model. Similar to that observed for global data, we observed varied performance in the training and testing phases. Furthermore, the qualitative performance seemed to improve by adopting the ensemble model. Furthermore, to qualitatively evaluate the model performances, the statistical parameters like standard deviation of error ($\sigma(\epsilon)$), Pearson correlation coefficient ($R$), Performance Parameter ($PP$), and Root Mean Squared Error ($RMSE$) values are obtained for

Western Himalayan model are presented in Table 5. In an ideal case, the value of ($R$) and ($PP$) should be unity, and the value of ($\sigma(\epsilon)$) should be zero. From Table 5, it is quite clear that Multilayer Perceptron (MLP) has performed best among the individual models or weak learners with Performance parameters (PP) value of 0.968 in the training and 0.685 in the testing phase. Also, an R-value of 0.989 in training and 0.848 in testing. The same was evident from Figure 12, where for both the training and testing phases, the MLP model is observed to capture the data variations better than other architectures. Furthermore, when the

prediction made from the individual techniques is employed as input for the ensembled random forest model, its performance increases significantly with RMSE values of 0.117 and 0.236 in the training and testing phases, respectively. It shows the satisfactory performance of the model, ensuring a reliable prediction. This improvement is also evident from the ensemble RF model presented in Figure 13, where the model is able to capture the peaks and troughs efficiently.

Furthermore, we attempted to forecast the expected annual seismic seismic energy release for the upcoming year (i.e. 2024).

Based on the model developed from this work, we can expect a total annual seismic energy in the range $9.11 \times 10^{14} J$ to $5.69 \times 10^{14} J$ Whereby, we can expect a maximum magnitude of 7.17 $M_w$ for the Western Himalayan region.





**Table 5.** The performance evaluation of individual models and the final ensemble random forest model at the training and testing phases for Western Himalayan region

| Models | Training | | | | Testing | | | |
|---|---|---|---|---|---|---|---|---|
| | $\sigma(\epsilon)$ | $R$ | $PP$ | $RMSE$ | $\sigma(\epsilon)$ | $R$ | $PP$ | $RMSE$ |
| MLP* | 0.180 | 0.989 | 0.968 | 0.221 | 0.659 | 0.848 | 0.685 | 0.687 |
| Linear Regression (LR) | 0.356 | 0.957 | 0.918 | 0.352 | 0.677 | 0.833 | 0.544 | 0.826 |
| Random Forest (RF) | 0.450 | 0.978 | 0.866 | 0.445 | 1.023 | 0.673 | 0.348 | 0.988 |
| SMOreg** | 0.475 | 0.924 | 0.835 | 0.492 | 0.574 | 0.903 | 0.688 | 0.684 |
| IBk*** | 1.088 | 0.447 | 0.164 | 1.090 | 1.133 | 0.425 | 0.180 | 1.108 |
| **Ensemble RF** | **0.114** | **0.996** | **0.990** | **0.117** | **0.235** | **0.991** | **0.963** | **0.236** |

\* MLP- Multi-Layer Perceptron; **SMOreg - Sequential Minimal Optimization regression;

***IBk - Instance-Bases learning with parameter k

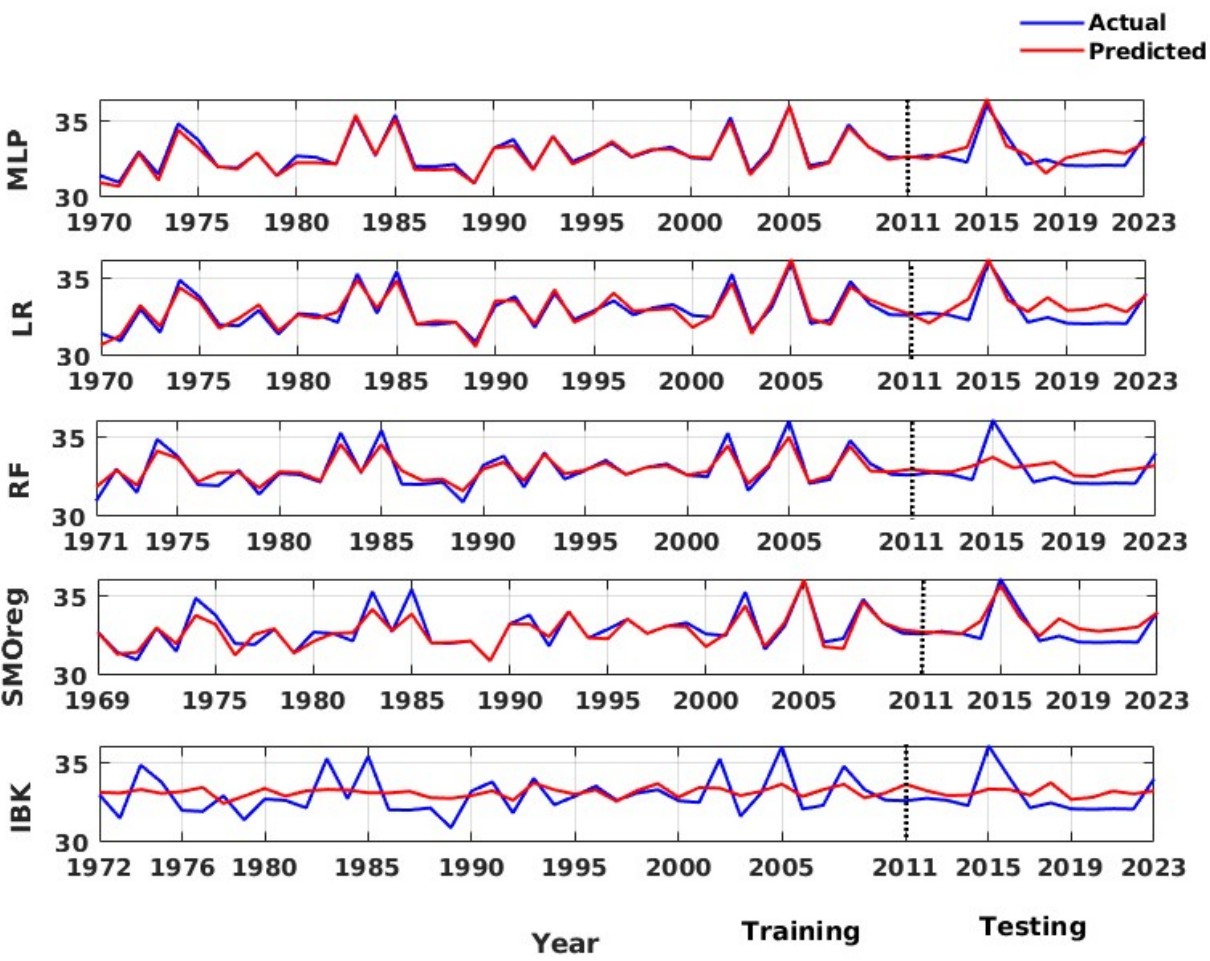

**Figure 12.** Actual vs Predicted values of log seismic energy from various individual techniques adopted for Western Himalayan region

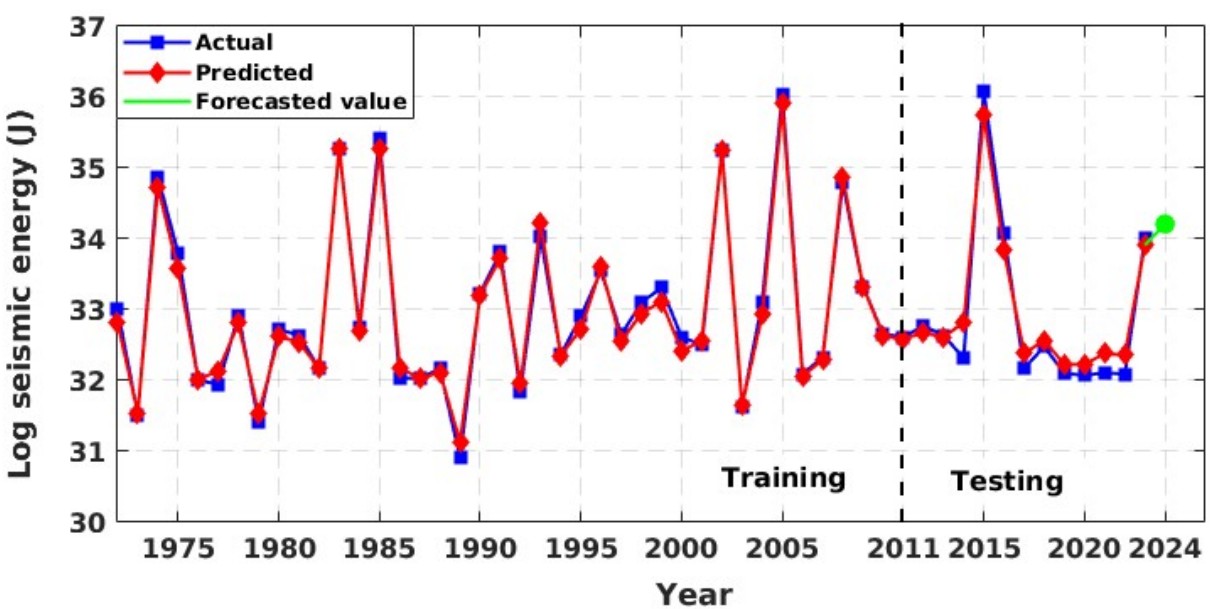

**Figure 13.** Actual vs Predicted values of seismic energy for Western Himalayan region from developed ensembled random forest model. Also the green marker shows the forecasted value for year 2024.





## 8   Discussion

This work investigated the application of sophisticated machine learning (ML) algorithms for seismic energy predictions. The proposed work is significant in quantifying the immediate hazard for the region. It is well known that seismic energy
is a potential indicator of seismic activity in the region. Thus, a reliable forecast through robust algorithms shall aid in the enhancement of hazard preparedness. Thus, a thorough modelling using five separate architectures representative of different structures in machine learning is attempted first. The model approaches considered are Multilayer Perceptron (MLP), Linear Regression (LR), Random Forest (RF), Sequential Minimal Optimization for Regression (SMOreg), and k-nearest Neighbors (IBk). Examining the separate models, it was found that MLP consistently performed better than the others during the training
and testing stages. The strong performance of MLP indicates that it might be a good fit for encapsulating intricate connections in seismic data. The model predictions from different architectures were observed to vary at the training and testing phases. Thus, the different ways in which these models function highlight how crucial it is to choose an algorithm that is suitable for the features of the seismic data. To improve the robustness of the prediction, the separate models (weak models) were combined to create an ensemble RF model. The outcomes showed that the performance of the ensemble model outperformed that of any
single model, highlighting the possible advantages of mixing several modelling techniques. Furthermore, when comparing the ensemble model to Raghukanth et al. (2017) model, the variance is lower, which suggests that the seismic energy forecasts are more stable and reliable. Even though the study used a worldwide time series that covered the years 1900 to 2015, it is important to recognize any potential limitations related to this temporal scope. It's possible that patterns of seismic activity change and that some recent occurrences go unrecorded. In order to overcome this constraint and maintain the predictive accuracy and relevance
of results, future research should train models on an updated catalog. The study's encouraging findings provide opportunities for more investigation. Thus, as a sample study, the regional-level forecast model is developed for the Western Himalayan region. As similar to the global model, the regional data performance in forecast improved while adopting an ensemble architecture. Even though the results are promising the analysis is done on a larger cluster combining 4 seismogenic zones in the region. A more detailed physics-based clustering and further application to forecast modelling is expected to provide more insights
into the spatiotemporal patterns of seismic activity. A more sophisticated knowledge of seismic energy trends may be obtained by extending the research to a more recent catalog and carrying out extensive regional-level investigations on regular basis. Furthermore, the potential to improve the accuracy of the forecast model through a rigorous feature selection approach can also be adopted. These shall be taken as the future scope of this work. Additionally, there are intriguing prospects to improve forecast accuracy further and capture complex patterns in seismic data by exploring more sophisticated and hybrid machine
learning techniques like Deep Learning, Extreme Learning Machine (ELM), and Generative Adversarial Networks (GANs). The benefits of using sophisticated machine learning techniques highlight the potential real-world uses for seismic energy prediction. The dependability of early warning systems is increased by increased forecast accuracy and decreased variability, which helps improve preparedness and mitigation tactics in seismically active areas.



# 9   Conclusions

The adoption of a suitable ensemble model is observed to significantly improve the model's accuracy towards forecasting seismic energy. Based on the confidence gained from testing with reported data and approach, the proposed approach was further employed to forecast seismic energy for the western Himalayas. Based on the developed model, we can expect a total annual seismic energy in the range $9.11 \times 10^{14} J$ to $5.69 \times 10^{14} J$, which is equivalent to a magnitude range of 7.03-7.17 in 2024. Thus we can expect a maximum magnitude of 7.17 $M_w$ for the Western Himalayan region. This study is a pilot study in 525 the direction of seismic energy release and forecast for the Himalayan region. A more detailed spatio-temporal investigation of the seismic energy patterns shall be taken up as the future scope of this work. Nevertheless, the study results and formulations are critical in hazard preparedness, immediate risk assessment and suitable policy formulations.

*Data availability.*   All the data is with corresponding authors and will be available on reasonable request.

*Author contributions.*   **Sukh Sagar Shukla:** Conceptualization, Data curation, Methodology, Validation, Visualization, Writing - original 530 draft. **J Dhanya:** Funding acquisition, Writing - original draft, Project administration, Supervision. **Priyanka:** Data curation, Formal analysis, Software. **Praveen kumar:** Data curation, Formal analysis, Software. **Varun Dutt:** Conceptualization, Resources, Software, Supervision.

*Competing interests.*   The authors declare that they have no known competing financial interests or personal relationships that could have appeared to influence the work reported in this paper.

# Funding

The authors would like to acknowledge seed grand at IIT Mandi to support this research under the project titled "Earthquake forecast and prediction model for Himalayas using machine learning approaches" with project number: IITM/SG/DJ/98.





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
