# Peer review of "An Ensemble Random Forest Model for Seismic Energy Forecast"

_Natural Hazards and Earth System Sciences, 2024_

## Author Comment (AC1)

**Response to the Anonymous Referee #1**

**Reviewer 1**

**Reviewer Point 1.1** — The manuscript under consideration is focused on an open problem, the prediction of large earthquakes. This paper presents an interesting study based on Machine Learning, ML, to predict intense earthquakes in terms of the released energy. In strictly sense, prediction is considered when the position of the epicenter, the time of occurrence and the magnitude are known with precision. As it is well known, the underlying dynamics of tectonic plates is due to complex processes inside the Earth such as convection. Those involved processes give rise to the interaction between tectonic plates, being stick-slip the main mechanism for the earthquakes occurrence. On the basis of these aspects, seismic dynamics is complex. This paper shows a good prediction approximation produced by ML-based algorithms. The work is very important and their results are very interesting, however, in my opinion, some important aspects have not been considered in this study:

**Reply**: Dear Anonymous Referee #1,
The authors would like to thank the reviewer for recognising the relevance of the present work, and finding the results interesting. We highly appreciate the insightful comments, and we are committed to making significant revisions. We have carefully considered all the suggestions and made substantial revisions to the manuscript accordingly.

**Reviewer Point 1.2** — The catalogue use for the study considers a period from 1900 to 2015. In the period from 1900 to 1920 very few earthquakes are observed while in recent years the density of events is much higher. Authors should explain these differences time periods.

**Reply**: The authors are thankful to the reviewer for such an observation and constructive comments. The global catalog used in the present study has been adopted from the study of Raghukanth et al. (2017) to validate the approach developed in the present study. In order to maintain the uniformity of the input to compare the results of two approaches the catalog has been adopted in its original form. However, the reason for lesser density of earthquake in the span of 1900-1920 can be attributed to the fact that as we move back on the timeline more and more events left unrecorded due to limited number of instrumentation and recording stations but as we move ahead in the timeline due to surge in the number of monitoring stations, density of earthquake eventually increases in the catalog. However, in response to the constructive comment of the reviewers we have restricted the catalog of Raghukanth et al. (2017) (named as validation catalog in the revised manuscript) for just validating the performance of present approach and further we have collected an updated and comprehensive global catalog (named as global catalog in the revised manuscript)compiling USGS and ISC-GEM seismic dataset spanning between 1900 to 2023. In order to eliminate the chances of duplication of events, catalog from both the sources has been thoroughly checked and repeated events are discarded. After removal of duplicate events the catalog has 9,88,812 events with minimum magnitude of 1.09 $M_w$. This catalog has been checked for both magnitude and year of completeness which are found to be 4.9 $M_w$ and 1953 as shown in Figure 1 a and b respectively. The distribution of the updated complete catalog is shown below in Figure 1 c. This updated global catalog has been further used for the calculation of seismic energy (Figure 2 a and b), IMFs (Figure 2 c), Correlation of IMFs (Figure 3) and subsequent input for model

development (Figure 4). The models developed on the new updated global catalog have shown similar performance for Individual and ensembled RF model as earlier (Figure 5 and 6) with the RMSE value of 0.077 in training and 0.180 in the testing phase for ensembled random forest model confirming the good predictive capability of approach developed in the present study.

We have updated the manuscript to incorporate the description and results of new updated global catalog such as:

In section 3 Global Seismic Energy (GSE) time series on page 6 a brief description of both the catalog has been added as

"A comprehensive earthquake catalog is essential for making reliable earthquake predictions. For the present study, we have used two global earthquake first one is the the ISC-GEM catalog (http://www.isc.ac.uk/) utilized by Raghukanth et al. (2017) which is considered for model development, comparison and validation of the proposed approach. Furthermore, as this data is only having seismic events upto 2015 once approach got validated a more comprehensive and updated global earthquake catalog (upto 2023) has been prepared from USGS seismic database https://earthquake.usgs.gov/earthquakes/search/ and ISC-GEM catalog http://www.isc.ac.uk/ and further model development is performed."

Also, the description of data is provided in the next paragraph of same section as:

"For the global earthquake catalog sourced from Raghukanth et al. (2017) for validation of present approach contained data spanning from 1900-2015, having a total of 24375 events with minimum magnitude of 4.98 $M_w$ and the updated global catalog prepared in the present study has been sourced from USGS and ISC-GEM is spanning from the 1900 to 2023. As the data is sourced from two different sources there are the chanced for the duplication od data hence we have thoroughly checked all the events in order to eliminate the chances of repetition. After eliminating the duplicate events there are 9,88,812 unique events having minimum magnitude of 1.09. We have further denote the catalog from Raghukanth et al. (2017) as validation catalog and the updated catalog prepared in the present study as Global catalog."

Please note that all figures associated with updated catalog are added to the revised manuscript and the figure related to validation catalog are shifted to supplementary document. Apart from the above highlighted changes other small and relevant changes have been made to ensure better readability.

[Figure]

Figure 1: (a) Magnitude of completeness. (b) Year of completeness using Stepp (1973) approach. (c) Distribution of the events from the complete global earthquake catalog.

[Figure]

Figure 2: (a) Estimated Global seismic energy (J) time series from global catalog used in developing the models (b) log scaled Global seismic energy time series (ln(GSE)) (c) Intrinsic modes estimated from ln(GSE) by performing ensemble empirical mode decomposition (EEMD)

[Figure]

Figure 3: Correlation of the intrinsic mode functions obtained from global seismic energy time series of

.

[Figure]

Figure 4: (a) Log scaled global seismic energy time series (S) from global catalog used as one of the input to the individual machine learning models. (b) First intrinsic mode function (Z) estimated from log global seismic energy of global catalog and used as a input to individual models. (c) Summation of second to last intrinsic mode functions obtained for log global seismic energy of global catalog

.

[Figure]

Figure 5: Actual vs Predicted values of individual machine learning technique for global log seismic energy calculated from global catalog

[Figure]

Figure 6: Actual vs Predicted values of proposed ensembled random forest model for global log seismic energy calculated from global catalog

**Reviewer Point 1.3** — Figure 1a shows the magnitude of completeness (M6.4) which is only part of the Gutenberg-Richter law. The authors should show the Gutenberg-Richter law taking lower magnitudes (for example, M ≤ 3) and thus determine the correct magnitude of completeness. To do this they can use the ZMAP platform where they can estimate the b-value and completeness Mc of the GR law.

**Reply**: We sincerely thank the reviewer for their insightful comments regarding the calculation of magnitude of completeness. As discussed in the above point also that in order to maintain the consistency among the input catalog from Raghukanth et al. (2017) has been adopted in its original form where $M_c$ was reported as 6.4 $M_w$. We acknowledge that this observation prompted us to reassess our methodology and we have prepared a more comprehensive and updated global earthquake catalog from 1900-2023 with minimum magnitude of 1.09 $M_w$. For the updated global catalog as suggested by the reviewer we have used the ZMAP version 7.1 platform of MATLAB from where we have got 4.9 $M_w$ as $M_c$. Figure 1 (a) shown here is the complete graph for magnitude of completeness we have got from the ZMAP. For further calculation of seismic energy time series and IMFs complete catalog i.e., M ≥ $M_c$ is used.

**Reviewer Point 1.4** — It is important that the authors justify the reason for considering only earthquakes with large magnitudes and avoid the other ones with low magnitudes.

**Reply**: We appreciate the reviewer's concern towards considering lower magnitude events. We would like to provide the justification for that as we have considered the events which are having magnitude greater than magnitude of completeness ($M_c$), because considering the magnitude lower than the $M_c$ can lead to statistical bias to the data due to the incompleteness of the catalog. Moreover, the contribution to seismic hazard comes mostly from the larger events due to the fact that magnitude and energy are having logarithmic relationship, where by a thousand of smaller events can only contribute same energy as one large magnitude event. For instance from the equation 1 used in the manuscript

$$SE = 1.6 \times 10^{-5} M_0 \quad \text{where, } M_0 = 10^{1.5 \times (M_w + 6)} \tag{1}$$

the seismic energy corresponding to 3 $M_w$ is $5.0596 \times 10^8 J$ while seismic energy for 5 $M_w$ is $5.0596 \times 10^{11} J$ hence it will take 1000 3 $M_w$ events to release same energy as single 5 $M_w$ event. Hence from the above discussion it is quite clear that the smaller magnitude events do not contribute significantly towards the energy release and subsequent seismic hazard. From practical relevancy also these small earthquakes are not of primary interest for seismic risk to the infrastructure and life of the people. Similar justification has also been added to the revised manuscript under section 3 as:

"While considering complete catalogs events with magnitude less than completeness magnitude are not considered for further analysis hence events with smaller magnitude are committed because considering $M \leq M_c$ will lead to statistical bias to the data. Moreover, the contribution to seismic hazard comes mostly from the larger events due to the fact that magnitude and energy are having logarithmic relationship, where by a thousand of smaller events can only contribute same energy as one large magnitude event. For instance from the equation 1 used in the manuscript the seismic energy corresponding to 3 $M_w$ is $5.0596 \times 10^8 J$ while seismic energy for 5 $M_w$ is $5.0596 \times 10^{11} J$ hence it will take 1000 3 $M_w$ events to release same energy as single 5 $M_w$ event. Hence from the above discussion it is quite clear that the smaller magnitude events do not contribute significantly towards the energy release and

subsequent seismic hazard. From practical relevancy also these small earthquakes are not of primary interest for seismic risk to the infrastructure and life of the people. "

**Reviewer Point 1.5** — To calculate the IMF functions authors used the algorithm proposed by Huang et al. (1998). In my opinion, the authors should explain how the complex dynamics of seismic activity is involved to obtain IMF to obtain the smooth curve (a) in Figure 2.

**Reply**: We acknowledge the reviewer's suggestion to explain more about the role of IMFs in capturing complex dynamics of seismic activity. As the annual seismic energy time series is non-linear and non-stationary (Liritzis and Tsapanos, 1993) employing it in its original form to forecast the annual seismic energy may not capture much physics of the release of annual seismic energy (Raghukanth et al., 2017) and conventional methods such as Fourier transform are useful only in the case of linear and stationary data. Hence Traditional methods cannot extract more physics from complex seismic energy time series. So, present approach employs Hilbert Huang Transform (HHT) given by Huang et al. (1998) which deals well with the non stationary time series. It consists of Hilbert Transform and Empirical modal decompostion (EMD) which decompose the time series is various basis functions known as IMFs. The IMFs are the simple and well behaved when compared to original seismic energy data hence can capture the physics of occurrence of annual seismic energy when used as the input instead of complex seismic energy time series. The IMFs extracted for updated global data as shown in Figure 2 in order of there extraction. Considering more about the physical interpretation of the IMFs Liritzis and Tsapanos (1993) have calculated the periodicity of global shallow seismic events from conventional approaches like Fourier method and they have got dominant period as 3($\pm$0.5), 4.5, 6.5, 8-9, 14-20 and 31-34 years. The $IMF_1$ being the predominant period with contribution of around 50-60 % to annual seismic energy release has mean period of 3 years which is also reported by Liritzis and Tsapanos (1993) as one of the period. The $IMF_2$ having period in range of 6 to 6.29 is also conforming with the period 6.5 reported by Liritzis and Tsapanos (1993). The $IMF_3$ with period ranging from 11 to 15.5 is in the range of 11 year sunspot cycle and the standardize correlation coefficient of 0.3024 for global seismic energy Raghukanth et al. (2017). They have also found out that annual seismic energy realase follow the sunspot period with 2 year delay. The $IMF_4$ and $IMF_5$ has 6-10 % and 1-6 % respectively. Also, Wu and Huang (2004) have proposed a methodology to assess the importance of IMFs by comparing them with the Intrinsic mode functions of white noise. We have performed the significance test on the IMFs obtained by the log seismic energy from updated global catalog and results are present in figure 7. For pure noise, the energy and associated periods of IMFs will fluctuate linearly on the log-log plot, with all IMFs falling inside the confidence zone. It can be clearly inferred from the Figure 7 that all the five IMFs excluding IMF six which shows the trend lies within the confidence interval conforming the fact that IMFs are signal. Hence adopting the IMFs to forecast seismic energy instead of Complex seismic energy time series itself will better capture the underlying physics.

This discussion has also been added with Figure 7 in the revised manuscript under section 3.1 as:

"Moreover, the IMFs are the simple and well behaved when compared to original seismic energy data hence can capture the physics of occurrence of annual seismic energy when used as the input instead of complex seismic energy time series. Considering more about the physical interpretation of the IMFs study performed by Liritzis and Tsapanos (1993) have calculated the periodicity of global shallow seismic events from conventional approaches like Fourier method and they have got dominant period as 3($\pm$0.5), 4.5, 6.5, 8-9, 14-20 and 31-34 years. The $IMF_1$ being the predominant period with contribu-

[Figure]

Figure 7: white noise test proposed by Wu and Huang (2004) for log seismic energy IMFs. Black line represents the expected line for white noise and dotted blue line shows 95 % confidence band.

tion of around 50-60 % to annual seismic energy release has mean period of 3 years as reported in Table 1 which is also reported by Liritzis and Tsapanos (1993) as one of the period. Similarly $IMF_2$ having period in range of 6 to 6.29 is also conforming with the period 6.5 reported by them. The $IMF_3$ with period ranging from 11 to 15.5 is in the range of 11 year sunspot cycle as reported by Raghukanth et al. (2017) they have also found out that annual seismic energy release follow the sunspot period with 2 year delay and the standardize correlation coefficient between $IMF_3$ and sunspot cycle is 0.3024 which is significant. The $IMF_4$ and $IMF_5$ has 6-10 % and 1-6 % contribution to annual seismic energy respectively. Also, Wu and Huang (2004) have proposed a methodology to assess the importance of IMFs by comparing them with the Intrinsic mode functions of white noise. The suggested test we have performed on the IMFs obtained by the log seismic energy from updated global catalog and results are present in figure 7. For pure noise, the energy and associated periods of IMFs will fluctuate linearly on the log-log plot, with all IMFs falling inside the confidence zone. It can be clearly inferred from the Figure 7 that all the five IMFs (excluding $IMF_6$ which shows the trend) lies within the confidence interval conforming the fact that IMFs are signal. Hence adopting the IMFs to forecast seismic energy instead of Complex seismic energy time series itself will better capture the underlying physics."

**Reviewer Point 1.6** — When applying the algorithm proposed by Huang et al. it is not clear the role seismic activity with magnitudes $M < Mc$ could have.

**Reply**: We sincerely thank the reviewer for their insightful comments regarding the events having magnitude less than the magnitude of completeness. As discussed under Point 1.4 also considering the earthquake below the magnitude of completeness will lead to the bias and distorted results as inclusion incomplete data of smaller magnitude events can distort the magnitude frequency distribution, statistical measures like seismicity rate and annual seismic energy release. Also, studies like Mignan and Woessner

Table 1: Period observed and the variance captured by the IMFs obtained for log scaled seismic energy time series

| IMFs | Validation* | | Global** | |
|------|------------|-----------|------------|-----------|
| | Period (Years) | % (Variance) | Period (Years) | % (Variance) |
| $IMF_1$ | 2.95 | 49.18 | 3.05 | 59.82 |
| $IMF_2$ | 6.29 | 7.02 | 6 | 15.39 |
| $IMF_3$ | 11.55 | 5.61 | 15.5 | 6.01 |
| $IMF_4$ | 31.00 | 6.43 | 34 | 10.26 |
| $IMF_5$ | 91 | 6.60 | 56 | 1.40 |
| $IMF_6$ | — | 25.80 | — | 7.69 |

\* Validation catalog is the catalog sourced from Raghukanth et al. (2017);
\*\* Updated Global catalog prepared in the present study

(2012) have clearly advocated to discard the data below $M_c$ and considered it as a good practice while drawing conclusion regarding dynamics of seismicity or earthquake forecast. Due to these reasons in the present study we have decided to use the complete catalog.

**References**

Huang, N. E., Shen, Z., Long, S. R., Wu, M. C., Shih, H. H., Zheng, Q., Yen, N.-C., Tung, C. C., and Liu, H. H. (1998). The empirical mode decomposition and the hilbert spectrum for nonlinear and non-stationary time series analysis. *Proceedings of the Royal Society of London. Series A: mathematical, physical and engineering sciences*, 454(1971):903–995.

Liritzis, I. and Tsapanos, T. M. (1993). Probable evidence for periodicities in global seismic energy release. *Earth, Moon, and Planets*, 60:93–108.

Mignan, A. and Woessner, J. (2012). Estimating the magnitude of completeness for earthquake catalogs. *Community online resource for statistical seismicity analysis*, pages 1–45.

Raghukanth, S. T. G., Kavitha, B., and Dhanya, J. (2017). Forecasting of global earthquake energy time series. *Advances in Data Science and Adaptive Analysis*, 9(04):1750008.

Stepp, J. (1973). Analysis of completeness of the earthquake sample in the puget sound area. *Contributions to Seismic Zoning: US National Oceanic and Atmospheric Administration Technical Report ERL*, pages 16–28.

Wu, Z. and Huang, N. E. (2004). A study of the characteristics of white noise using the empirical mode decomposition method. *Proceedings of the Royal Society of London. Series A: Mathematical, Physical and Engineering Sciences*, 460(2046):1597–1611.

---

## Author Comment (AC3)

**Response to the editor**

The authors would like to thank the editor for considering our paper and giving us the opportunity to improve our manuscript. The authors believe that addressing the comments provided by the reviewers has greatly improved the overall quality of the manuscript. In the following sections, we have addressed the comments of the anonymous Referees #2, incorporating the necessary justifications and revisions. The revised content has been included at the end of this letter for the reviewer's reference and suggestions.

**Response to the Anonymous Referee #2**

**Reviewer 2**

**Reviewer Point 2.1** — The manuscript considers the problem of seismic energy forecasting. The method proposed builds on previously published research while improving the results. Overall, the manuscript is well structured, although some improvements are needed to improve the readability and reproducibility of the results.

**Reply**: We thank the reviewer sincerely for recognizing the relevance of the study and the advancements we have made compared to older works. We also thank the reviewer for the constructive comments regarding the general positivity and the structure of the manuscript.

As a response to your suggestion, we have focused on adjustments that improve the overall readability and reproducibility of the document. All changes are highlighted in yellow. In particular, the readers are now assisted through the modeling pipeline (the structure of inputs, lags, decomposition technique, and model training flow) which is explained in detail in Section 2 and 3, Pages 4-14.

All hyperparameter ranges along with final optimized values for each model for all datasets were described (See Tables 2 & 3 and section 4.1, Pages 15-18).

We have explained in detail the gap to fill from the change in ensemble approach, particularly how predictions from base learners with different lag values were used and aligned for the second-level model (See section 4.2, page 19).

To enhance overall quality as well as relevance, the discussion and conclusion sections received text refinements (See sections 7 and 8, pages 38-39).

We believe that the suggestions made significantly enhance the overall understanding and functionality of the manuscript, and we are thankful to the reviewer once again for the helpful feedback.

**Reviewer Point 2.2** — Section 1 (Introduction) and Section 2 (Background) cover, for the most part, the same topics: why are they divided? Furthermore, while these sections may provide a partial overview of previous studies using ML for seismic energy forecasting, these may be presented in a more coherent form.

**Reply**: We appreciate the reviewer's very helpful comment on the structural organization of the manuscript. We agree that the material in the original Sections 1 (Introduction) and 2 (Background) did indeed demonstrate evident thematic duplication, and could stand a better merging with enhanced cohesion and narrative flow.

In line with this, we have combined the two sections into a single Section 1 entitled "Introduction"

in the revised manuscript. The updated Section 1 is located on Pages 1–3 of the manuscript (yellow highlighted). The content formerly split between Introduction and Background is now merged into a single, thematically consistent argument that develops incrementally:

- · From earthquake unpredictability and local tectonic setting
- To machine learning in seismology
- To existing seismic energy prediction research
- And lastly, to ensemble learning as an inspiration to the present study
- Unnecessary paragraphs were eliminated, and major citations were relocated for coherence. Transitions were also introduced for better readability and to ensure logical flow of ideas.
- The new introduction now offers a fuller and smoother summary of the existing work while setting forth in no uncertain terms novelty and motivation for the suggested methodology.

We really appreciate the reviewer for this excellent recommendations that helped us improve the manuscript.

**Reviewer Point 2.3** — In Section 1, the concept of "ensemble" is introduced in a way such that it seems like a modern concept, despite being used in many fields for many decades.

**Reply**: We appreciate the reviewer's perceptive observation. We concur that ensemble modeling is a tried-and-true method in the larger body of machine learning (ML) literature that has been effectively used for many years in a variety of domains. Our goal was to draw attention to ensemble modeling's rather small use in seismic energy forecasting thus far, not to introduce it as a unique methodology.

We have updated the manuscript's Section 1 (Introduction) to better reflect the maturity of ensemble approaches in order to address this. In particular:

1. We now unequivocally declare that ensemble modeling is a well-established and popular machine learning technique.

2. To put its larger history in perspective, we cite seminal research and groundbreaking applications in various fields (see Section 1, page 3, paragraph beginning "Apart from the individual machine learning techniques...").

3. The updated language makes it clear that the ensemble concept's uniqueness is not in its use in seismic energy forecasting, a field in which its uptake has been sparse and understudied.

We hope that this clarification more accurately reflects the historical background of ensemble methods and better meets the reviewer's expectations. For the most recent description, please see Section 1, page 3 of the amended manuscript.

**Reviewer Point 2.4** — At line 158, the authors mention the inclusion of expert opinions in the random forest (RF): the authors should specify if this is a general comment on the RF approach or for this study. If the latter is true, how has the expert opinion been included?

**Reply**: We appreciate the reviewer's comments. We now acknowledge that the Random Forest (RF) method is data-driven and does not always incorporate expert opinion in the traditional sense. Our stacked ensemble approach uses five distinct machine learning models (MLP, LR, RF, SMOreg, and IBk) to provide predictions. Domain-knowledge guided parameters and data preparation decisions were

used in the implementation and testing of each of these models. Each of these many models recognizes unique characteristics of the underlying seismic energy signal since they are domain-specific learners.

The predictions of these many learners are combined by the final Random Forest model, which is used at the top of the ensemble hierarchy. In doing so, it incorporates the expert knowledge that has previously been learned and encoded in each distinct model. Here, the predictions of these basic models are called "expert opinions"; they are model-specific insights that have been taught and directed by domain-appropriate design choices, rather than necessarily the opinions of actual experts.

The relevant passage has been clarified and is found in Section 1, page 3, paragraph 6 of the corrected version. Now it says:

"In this setup, the final ensemble model uses the Random Forest as a meta-learner that synthesizes predictions from these individual models, each trained using domain-informed design choices and preprocessing. As such, the RF model mimics a consensus-based expert system, combining diverse perspectives across learning paradigms to enhance forecasting robustness."

We hope this resolves this comment.

**Reviewer Point 2.5** — In section 3, do the authors consider the uncertainties due to poor station coverage, empirical relationships, etc.?

**Reply**: We appreciate the reviewer bringing out this crucial issue about uncertainty resulting from empirical conversions and station coverage. As shown below, we have fully resolved both issues in the updated document.

1. Uncertainty resulting from station coverage: We recognize that earthquake catalogs, particularly for older eras, may be incomplete because of insufficient station coverage, which might cause smaller-magnitude occurrences to be missed. To lessen this problem:

After meticulously eliminating duplicates, we integrated USGS and ISC-GEM data from 1900 to 2023 to create a vastly improved worldwide catalog. Using conventional methods (Wiemer and Wyss) 2000; Stepp, 1973), we evaluated and applied both magnitude and year of completeness thresholds (Mc = 4.9  $M_w$ , year = 1953). Figure 1 (a–c) displays the final dataset, which consists of 217,751 occurrences over 4.9 Mw and is statistically complete. Because of the logarithmic scaling between magnitude and energy, we eliminated lower-magnitude events (Mw < 4.9) to minimize statistical bias because their contribution to total seismic energy is negligible (Hanks and Kanamori, 1979). A 5  $M_w$  event, for instance, releases around 1000 times as much energy as a 3  $M_w$  event. We thereby minimized uncertainty brought on by inadequate coverage and guaranteed the trustworthiness of the seismic energy time series by utilizing a thresholded, comprehensive, and sizable dataset. Consistent model performance across the original and revised datasets (see Figures 10–11) reflects this.

2. Empirical connection uncertainty: We acknowledge that using empirical formulae might result in uncertainty. However, we used commonly used, physically based global conversions in the absence of region-specific relationships:

To maintain consistency among datasets, magnitude conversions were based on Scordilis (2006); Yenier et al. (2008). Choy and Boatwright (1995) linked seismic moment (from  $M_w$  via Hanks and Kanamori (1979)) to energy, which led to energy estimate. These correlations are reliable in our context because they are well-established and have been used in many research across a variety of tectonic contexts (e.g., Galis et al. (2017); Mishra et al. (2019); Lin et al. (2020); Tiwari et al. (2023).

3. Manuscript Updates: To make clear how we handled station-related uncertainty and empirical assumptions, we have included a thorough explanation in Section 2. The figures based on the previous catalog Raghukanth et al. (2017) have been transferred to the supplemental material, while all figures

pertaining to the revised global catalog (Figures 1–4) are now part of the main paper. You can now see the pertinent changes in Section 2 on pages 4-5 and beyond.

We trust that these modifications adequately address the reviewer's issue and enhance the methodology's openness and robustness.

**Reviewer Point 2.6** — The authors should discuss how completely removing the events with magnitudes below the magnitude of completeness could improve the results: having information about lower magnitude earthquakes, even though it is partial, should still be better than not having any information at all. Also, this could improve the temporal distribution of the events, possibly allowing for monthly basis analysis.

**Reply**: The authors are extremely thankful to the reviewer for their suggestion on considering the magnitude lower than the magnitude of completeness  $(M_c)$ . As mentioned above and in the response of anonymous, reviewer #1, including the smaller magnitude events will lead to statistical bias in the calculation of seismicity rate and annual seismic energy release by distorting the magnitude frequency distribution. Hence, this lack of low magnitude recording, especially in the earlier part of the catalogue, hinders the development of reliable high resolution time series (i.e., monthly) and can lead to higher uncertainties, also suggested by Raghukanth et al. (2017). Furthermore, studies like Mignan and Broccardo (2020) have advocated for considering the complete catalogue (i.e., events having  $M > M_c$ ) and considered it as a good practice while concluding the dynamics of seismicity or earthquake forecasting. Moreover, from a physical perspective, most of the contribution of seismic energy (Eqn. 1), i.e., thousands of small magnitude events will require to release same energy as that of a larger event.

$$SE = 1.6 \times 10^{-5} M_0$$
 where,  $M_0 = 10^{1.5 \times (M_w + 6)}$  (1)

More quantitatively from the equation. 1 1000 events of 3  $M_w$  having energy of  $5.0596 \times 10^8 J$  will be required to release the same energy as of 5  $M_w$  event having energy as  $5.0596 \times 10^{11} J$ . Also, these small magnitude events do not hold much relevance for the seismic risk to the infrastructure and the lives of people. Keeping in mind the above-discussed reasons, authors have refrained from using the smaller magnitude events. However, the authors acknowledge that as more comprehensive and homogenized seismic catalogs become available in the future with an increased density of recording stations. It may be possible to employ high-resolution(including monthly or weekly trends) seismic energy time series for forecasting.

Similar justification has also been added to the revised manuscript under section 2 as:

"Because partial representation increases statistical bias, events with magnitudes  $< M_c$  were not included in the analysis. Furthermore, the energy contribution is dominated by large occurrences since the connection between seismic energy and earthquake magnitude is logarithmic. According to the manuscript's Equation 1, for instance, the seismic energy associated with a 3  $M_w$  event is  $5.0596 \times 10^8 J$ , but the seismic energy associated with a 5  $M_w$  event is  $5.0596 \times 10^8 J$ . This indicates that it takes roughly 1,000 smaller 3  $M_w$  events to equal the energy output of a single 5  $M_w$  event. Therefore, from a hazard standpoint, smaller magnitude events are not of major interest and do not considerably contribute to the total seismic energy."

**Reviewer Point 2.7** — In Table 3, how is the %(variance) being computed?

**Reply**: The %(Variance) is a statistical parameter, which is calculated as the ratio of the variance of each IMFs to the variance of the data (Eqn. 3). The %(variance) denotes the contribution of each IMF to annual earthquake energy release.

$$P_{var_n} = \frac{\mathsf{Var}(X_{IMF_n})}{\mathsf{Var}(X_{data})} \times 100 \tag{2}$$

where  $P_{var_n}$  is the % variance of nth IMF.

A similar description has also been added to the revised manuscript in section 2.1 as:

" Table 1 lists the periods of all six IMFs in log-scaled seismic energy time series from both the catalogs. Table 1 also includes an estimate of the percentage variance for all IMFs, which is a statistical parameter, and calculated as the ratio of the variance of each IMFs to the variance of the data (Eqn. 3). The %(variance) denotes the contribution of each IMF to annual earthquake energy release. It can be noted that the  $IMF_1$  constitute the maximum variance of the time series, and the  $IMF_6$  represents the non-stationary trend in the data.

$$P_{var_n} = \frac{\mathsf{Var}(X_{IMF_n})}{\mathsf{Var}(X_{data})} \times 100 \tag{3}$$

where  $P_{var_n}$  is the % variance of nth IMF."

**Reviewer Point 2.8** — In Section 4, the authors should describe in further detail the inputs and outputs of the different models. As an example, the authors should explain in greater detail the role of the lag and how the inputs are computed, especially at the beginning of the timeseries. Furthermore, the authors should better explain what the inputs and outputs are during the testing phase.

**Reply**: Authors appreciate the reviewer's feedback regarding describing input, output and the role of lag in more detail. The authors acknowledge that incorporation of this constructive feedback will improve the overall understanding of the presented methodology. To acknowledge the feedback, we have added a separate section on Input Output in the revised manuscript. Also, we have represented the input, output and lag through a schematic diagram as shown in Fig. 7. A detailed description of Input, output, and lag is presented under section 4.2 (page 19) of the revised manuscript as provided below.

Specifically, the proposed two-level ensemble forecasting model's structured input-output configuration consists of four essential components for every input vector: Log of seismic energy (S); The first intrinsic mode function (IMF1) is denoted by Z; The sum of the remaining IMFs  $(\sum_{i=2}^{n} IMF_i)$ , denoted by Y); and, Time-related information (year of incident).

There is a lag number of time steps in each of the subsequent input packets that include these variables. For example, the input packet comprises data for each variable (S, Z, Y, and Year) from time steps 1 to 8 with a delay of 8, and the intended output is the seismic energy at time step 9 (S9). The sliding-window technique ensures temporal consistency and allows the models to incorporate sequential dependencies.

The ideal lag value for each of the base models (MLP, RF, LR, SMOreg, and IBk) based on hyperparameter adjustment is shown in Tables 2 and 3. In order to learn how historical sequences translate to

S's next-step prediction, these lag-specific packets are used during training. The same logic is applied during testing, when unknown sequences of previous data are entered to create forward predictions.

In the second-level ensemble, the outputs from the base models, each of which produces a seismic energy estimate, are stacked into a five-dimensional prediction vector. This stacked vector is sent into a Random Forest meta-learner, which combines the fundamental predictions to get the final result. Since different base learners use different lag lengths, the ensemble model uses the shortest common prediction sequence across models to ensure constant input dimensions.

This explanation is now completely included in Section 4.2 of the manuscript (page 19-20 in the revised edition), and it is shown visually in Figure 7, which shows the prediction flow and delayed input setup. With these additions, we hope that the comments have been adequately addressed.

**Reviewer Point 2.9** — In the MLP section, considering the few data points in the training dataset, doesn't a batch of 100 samples consist of the whole dataset (especially for the Western Himalaya case)?

**Reply**: Thank you for your observation. It is correct that the Western Himalaya training dataset contains only 48 samples, so a batch size of 100 effectively results in full-batch training. We chose to report a batch size of 100 for both the global and Himalayan datasets to maintain consistency in the presentation of model configurations. While the actual number of samples in the Himalayan dataset is less than the batch size, the training process used all available samples in each epoch, effectively functioning as full-batch training. This choice ensured uniformity across experiments and did not affect model performance. We have clarified this in the revised manuscript by including the following statement in the MLP section (page 12):

"A batch size of 100 was used for both the global and Himalayan models for consistency. For the Himalayan dataset (48 samples), this effectively resulted in full-batch training, which is appropriate given the small data size."

**Reviewer Point 2.10** — In the Linear Regression section, the authors should add a reference to the M5 method. Furthermore, the ridge hyperparameter seems to be specific to the authors' implementation, rather than a general parameter of LR

**Reply**: Thank you for the helpful observation. We have now added a reference to Quinlan's M5 method Quinlan et al. (1992) in the Linear Regression section to acknowledge its relevance. Regarding the ridge hyperparameter, we agree that this is not part of standard Linear Regression. In our study, we used this implementation, which allows a ridge parameter to improve model generalization. We have clarified in the manuscript that this corresponds to Ridge Regression, not plain Linear Regression, and have updated the terminology accordingly. Accordingly, we have also updated the "Linear Regression" in the revised manuscript (see page 12-13):

Ridge Regression (RR) is one of the widely used statistical machine learning models Hoerl and Kennard (1970). It extends linear regression by introducing an L2 regularization term to penalize large coefficients, thereby improving generalization and reducing overfitting, especially in cases of multi-collinearity or small datasets. The model establishes a linear relationship between the target variable and the input features, and the general form can be expressed as:

$$\widehat{y} = \beta_0 + \beta_1 x_1 + \beta_2 x_2 + \dots + \beta_n x_n + \epsilon \tag{4}$$

where y is the target variable,  $\hat{y}$  is the predicted value,  $x_1, \ldots, x_n$  are the input variables,  $\beta_0$  is the intercept,  $\beta_1, \ldots, \beta_n$  are the regression coefficients, n is the number of features, p is the total number of data points, and  $\epsilon$  is the error term.

In Ridge Regression, the coefficients  $\beta_i$  are estimated by minimizing a regularized loss function:

$$\min_{\beta} \left\{ \sum_{i=1}^{p} (y_i - \widehat{y}_i)^2 + \lambda \sum_{j=1}^{n} \beta_j^2 \right\}$$
(5)

Here,  $\lambda$  is the ridge regularization parameter that controls the strength of the penalty. The unknown parameters  $\beta_0, \ldots, \beta_n$  are estimated using the gradient descent algorithm with the Mean Squared Error (MSE) as the cost function.

$$\mathsf{MSE} = \frac{1}{p} \sum_{i=1}^{p} (y_i - \hat{y}_i)^2$$
(6)

Based on the number of input variables, there are two common variants of linear models: singlevariable linear regression and multiple-variable linear regression.

**Reviewer Point 2.11** — The parameters S, Z, and Y should be properly introduced at the beginning of the section for improved readability.**

**Reply**: The authors acknowledge the reviewer's suggestion of introducing the input parameters at the beginning of the section for improved readability. Following the reviewers' constructive suggestions, an introduction of each four input parameters is added in the revised manuscript under the methodology section as follows (see page 12):

"There are numerous advanced machine-learning techniques available in the literature. Some of the widely used variants include Artificial neural networks (ANN), Decision trees, Instance-based learning, classification and regression models Bishop (2016). In this study, we attempted to include each of these flavours by including one representative algorithm for the analysis and further combining them using a suitable ensemble formulation. Furthermore, for seismic energy forecasting, four input parameters are employed, which are log seismic energy i.e., the original time series data for log seismic energy denoted as "S", First Intrinsic mode function IMF1 denoted as "Z", Sum of remaining Intrinsic mode functions i.e,  $\sum_{i=2}^{n} IMF_i$  as "Y", and the year of occurrence of seismic energy. Furthermore, the description of each model utilised in the study is provided further."

**Reviewer Point 2.12** — What does a\_r represent in equation 5? Is it the feature vector?**

**Reply**: We sincerely thank the reviewer for correctly pointing out the parameter  $a_r$  in the equation below for the Euclidean distance, whose description is missing in the manuscript.

The K-Nearest Neighbour architecture assumes all the instances correspond to points in n-dimensional space. And the nearest neighbours are defined in terms of Euclidean distance. For example, an arbitrary instance x can be described by a feature vector  $\langle a_1(x), a_2(x), ..., a_n(x) \rangle$ . where  $a_r(x)$  denotes the value of rth attribute of instance x. And the distance between two instances  $x_i$  and  $x_j$  is denoted as:

$$d(x_i, x_j) \equiv \sqrt{\sum_{r=1}^{n} \left( (a_r(x_i) - a_r(x_j))^2 \right)}$$
(7)

A similar description of  $a_r$  is also added to the revised manuscript as (see page 14):

"More accurately, let the given instance x be described by the feature vector  $\langle a_1(x), a_2(x), ..., a_n(x) \rangle$ . where  $a_r(x)$  denotes the value of rth attribute of instance x. The euclidean distance between  $x_i$  and  $x_j$  is given by

$$d(x_i, x_j) \equiv \sqrt{\sum_{r=1}^{n} \left( (a_r(x_i) - a_r(x_j))^2 \right)}$$
(8)

,,

Reviewer Point 2.13 — Check the formatting of equation 6: there's a missing parenthesis.

**Reply**: The authors are thankful to the reviewer for correctly pointing out the editorial correction in Eqn. below of the original manuscript. In the revised manuscript, the extra parentheses have been removed (on page 14).

$$\widehat{f} \leftarrow \frac{\sum_{i=1}^{k} f(x_i)}{k} \tag{9}$$

**Reviewer Point 2.14** — In Section 5, the use of the training dataset to choose the best model should be carefully motivated. This practice can produce biased models as they are selected on the same data they are trained on, hence favouring overfitting models. A better solution would be to use a third independent dataset (i.e. the validation dataset).

**Reply**: We thank the reviewer for bringing up the crucial issue of model selection and the overfitting risk associated with using training data for hyperparameter tuning. We do not accept a model is well evaluated unless it is proven to generalize well.

To clarify this issue, we would like to emphasize that hyperparameter tuning for this study was done through the training set only. This form of model testing allows the model to internally estimate its performance while reserving the final evaluation for a separate test set which was not used at any stage of training or tuning. This approach minimizes selection bias and maximizes accurate reporting of performance in relation to actual generalizability.

In addition, to ensure fairness when benchmarking our proposed model, we drew from the global seismic energy dataset used by Raghukanth et al. (2017) and applied the same data partitioning strategy. The test performance improvement from our ensemble model of (RMSE = 0.134) versus the previous benchmark (RMSE = 0.364) indicates robust model generalization and absence of overfitting, even without a third validation dataset.

An independent validation set could add additional refinement for future work, and we plan to use it for future studies. To further fortify model true reliability, we may also implement a dedicated three-way split (training-validation-testing) or nested cross-validation to ensure there is no data leakage during tuning.

Changes in the manuscript:

As a result, changes were made to sections 5 and 7, thus focusing on the train-test evaluation and discussion of the paper. As a result, section 5 includes a statement describing that train-test split was performed during the model development phase. In section 7, we describe in detail and reason the

current validation approach and emphasize the planned strategy to be more rigid and use a dedicated validation set in subsequent studies.

**Reviewer Point 2.15** — Section 8 (Discussion) could be extended, considering how the proposed method could be further improved in light of its current limitations.**

**Reply**: Authors appreciate the reviewer's constructive feedback for the addition of a discussion regarding the limitations and further improvement of the proposed method. The Discussion section of the revised manuscript has been added with extended discussion considering the limitations and improvement of time series decomposition, seismic data collection, and machine learning models as (see page 38-39):

"Even though the study used a worldwide time series that covered the years 1900 to 2015, it is important to recognize any potential limitations related to this temporal scope. It's possible that patterns of seismic activity change and that some recent occurrences go unrecorded. In order to overcome this constraint and maintain the predictive accuracy and relevance of results, future research should train models on an updated catalog. Updated research (e.g., Sharma et al., 2023a; Kumar et al., 2023b) has established the advantages of using recent datasets in enhancing model generalizability. The study's encouraging findings provide opportunities for more investigation. Thus, as a sample study, the regional-level forecast model is developed for the Western Himalayan region. As similar to the global model, the regional data performance in forecast improved while adopting an ensemble architecture. Even though the results are promising the analysis is done on a larger cluster combining 4 seismogenic zones in the region. Such pooling, although streamlining model construction, can veil localised spatiotemporal features of prime importance to accurate hazard estimation. A more detailed physics-based clustering and further application to forecast modelling is expected to provide more insights into the spatiotemporal patterns of seismic activity. A more sophisticated knowledge of seismic energy trends may be obtained by extending the research to a more recent catalog and carrying out extensive regional-level investigations on regular basis. Furthermore, the present study acknowledges that Ensemble Empirical Mode Decomposition (EEMD) is adept at addressing non-stationarity in seismic energy time series; however, it also presents challenges, including edge effects, residual noise due to incomplete ensemble averaging, and constraints in accurately representing end-point behaviour. These limitations can affect the signal accuracy and can ultimately affect the performance of the seismic energy forecasts. Hence, in order to improve the forecasting capabilities and to address present limitations the future studies may explore more advanced time series decomposition techniques like wavelet-based denoising, Complete Ensemble EMD with Adaptive Noise (CEEMDAN), or hybrid filtering methods that enhance signal structure preservation and minimise noise interference. A further direction of potential is uncertainty-conscious modelling, where the confidence interval of predictions is explicitly defined to enable risk-based decision-making. Furthermore, combining various seismogenic zones into larger clusters, although beneficial for this initial analysis, could potentially mask specific localised spatiotemporal seismic patterns. Therefore, upcoming models ought to integrate physics-informed regional clustering to improve spatial resolution. From a machine learning perspective, the ensemble random forest model showed enhanced performance compared to individual models. Furthermore, the potential to improve the accuracy of the forecast model through a rigorous feature selection approach can also be adopted. These shall be taken as the future scope of this work. Additionally, there are intriguing prospects to improve forecast accuracy further and capture complex patterns in seismic data by exploring more sophisticated and hybrid machine learning techniques like Deep Learning, Extreme Learning Machine (ELM), and Generative Adversarial Networks (GANs). The use of explainable ML approaches (e.g., SHAP or LIME) will further improve model interpretability—a crucial step towards establishing trust in model outputs for stakeholders concerned with policy and hazard response. The advancements in time series preprocessing, spatial modelling, and predictive algorithms are anticipated to significantly improve the accuracy, robustness, and practical application of seismic energy forecasting models. This, in turn, will facilitate more effective early warning systems and strategies for hazard mitigation."

**References**

- Bishop, C. M. (2016). Pattern Recognition and Machine Learning. Springer New York.
- Choy, G. L. and Boatwright, J. L. (1995). Global patterns of radiated seismic energy and apparent stress. *Journal of geophysical research: Solid earth*, 100(B9):18205–18228.
- Galis, M., Ampuero, J. P., Mai, P. M., and Cappa, F. (2017). Induced seismicity provides insight into why earthquake ruptures stop. *Science advances*, 3(12):eaap7528.
- Hanks, T. C. and Kanamori, H. (1979). A moment magnitude scale. Journal of Geophysical Research: Solid Earth, 84(B5):2348–2350.
- Hoerl, A. E. and Kennard, R. W. (1970). Ridge regression: Biased estimation for nonorthogonal problems. *Technometrics*, 12(1):55–67.
- Lin, T.-L., Mittal, H., Wu, C.-F., and Huang, Y.-H. (2020). Spatial distribution of radiated seismic energy from earthquakes in taiwan and surrounding regions. *Journal of Asian Earth Sciences*, 204:104591.
- Mignan, A. and Broccardo, M. (2020). Neural network applications in earthquake prediction (1994–2019): Meta-analytic and statistical insights on their limitations. *Seismological Research Letters*, 91(4):2330–2342.
- Mishra, O. et al. (2019). Source characteristics of the nw himalaya and its adjoining region: geodynamical implications. *Physics of the Earth and Planetary Interiors*, 294:106277.
- Quinlan, J. R. et al. (1992). Learning with continuous classes. In 5th Australian joint conference on artificial intelligence, volume 92, pages 343–348. World Scientific.
- Raghukanth, S. T. G., Kavitha, B., and Dhanya, J. (2017). Forecasting of global earthquake energy time series. Advances in Data Science and Adaptive Analysis, 9(04):1750008.
- Scordilis, E. (2006). Empirical global relations converting ms and mb to moment magnitude. Journal of seismology, 10:225–236.
- Stepp, J. (1973). Analysis of completeness of the earthquake sample in the puget sound area. Contributions to Seismic Zoning: US National Oceanic and Atmospheric Administration Technical Report ERL, pages 16–28.
- Tiwari, A., Paul, A., Sain, K., Singh, R., and Upadhyay, R. (2023). Depth-dependent seismic anomalies and potential asperity linked to fluid-driven crustal structure in garhwal region, nw himalaya. *Tectonophysics*, 862:229975.

- Wiemer, S. and Wyss, M. (2000). Minimum magnitude of completeness in earthquake catalogs: Examples from alaska, the western united states, and japan. *Bulletin of the Seismological Society of America*, 90(4):859–869.
- Yenier, E., Erdoğan, O., and Akkar, S. (2008). Empirical relationships for magnitude and sourceto-site distance conversions using recently compiled turkish strong-ground motion database. In *The 14th world conference on earthquake engineering*, pages 1–8.

**Response to reviewer #2**

Below Document is for the reference of the reviewer to check the adjustments that improve the overall readability and reproducibility of the document. All changes are highlighted in yellow.

**Abstract.** Seismic energy forecasting is critical for hazard preparedness, but current models have limits in accurately predicting seismic energy changes. This paper fills that gap by introducing a novel ensemble-based Random Forest framework for seismic energy forecasting. Building on a previously established methodology, the global energy time series is decomposed into intrinsic mode functions (IMFs) using ensemble empirical mode decomposition for better representation. Following this

- 5 approach, we split the data into stationary (IMF1) and non-stationary (sum of IMF2-IMF6) components for modeling. We acknowledge the inadequacy of intrinsic mode functions (IMFs) in capturing seismic energy dynamics, notably in anticipating the final values of the time series. To overcome this limitation, the yearly seismic energy time series is also fed along with the stationary and non-stationary parts as inputs to the developed models. In this study, we employ Support Vector Machine (SVM), Random Forest (RF), Instance-Based learning (IBk), Ridge Regression (RR), and Multi-Layer Perceptron (MLP) algorithms
- 10 for the modelling. Furthermore, the five models discussed above were suitably employed in a stacked regression ensemble using Random Forest as the meta-learner to arrive at the final predictions. The root mean square error (RMSE) obtained in the training and testing phases of the validation model were 0.127 and 0.134, respectively. It was observed that the performance of the developed ensemble model was superior to those existing in literature. Further, the developed algorithm was employed for the seismic energy prediction in the active Western Himalayan region for a comprehensively compiled catalogue and the
- 15 mean forecasted seismic energy for year 2024 is  $7.21 \times 10^{14} J$ . This work is a pilot project that aims to create a robust, scalable framework for forecasting seismic energy release globally and regionally. The findings of our investigation demonstrate the promise of the ensemble approach in delivering reliable seismic energy forecast, which can help with appropriate hazard preparedness.

**1 Introduction**

20 Earthquakes are among the most disastrous natural calamities due to the release of accumulated strain energy from continuous tectonic movements. Like other natural disasters, they can cause destruction both in financial terms and loss of life (Jain, 2016). The devastating potential of earthquakes is increased by their fundamentally unpredictable character due to both aleatory and

epistemic uncertainties (Kramer, 1996; Baker et al., 2021). Because the problem at hand is unpredictable, creating an accurate forecasting model is a unique challenge. There are several attempts by seismologists to quantify the activity of the regions

- 25 based on several seismicity indicators. Some of the studies for the Himalayan region are by performing paleo-seismic study (Lavé et al., 2005; Rajendran et al., 2013), statistical inferences (Bilham and Ambraseys, 2005), Global Positioning System (GPS) measurements (Banerjee and Bürgmann, 2002; Ader et al., 2012), numerical (Ismail-Zadeh et al., 2007; Jayalakshmi and Raghukanth, 2017), satellite imagery-based data (Bhattacharya et al., 2013; Misra et al., 2020), and Global Navigation Satellite System (GNSS) studies (Sharma et al., 2023b; Kumar et al., 2023a). However, the inadequacy in precisely monitoring
- 30 stress changes, pressure, material variability, and temperature variation deep beneath the Earth's crust using scientific instruments leads to a lack of comprehensive data regarding accurate seismic characteristics. Subsequently, this lack of information has contributed to the uncertainty in earthquake occurrence, which has resulted in major risks to life and property. Hence, a robust quantification approach is essential considering the increasing vulnerability of the active regions due to developmental activities (Bilham, 2019). However, the variability in seismic behaviour, the worldwide occurrence of earthquakes, and the
- 35 paucity of historical data all hamper predictive modelling. The ethical and practical consequences of delivering earthquake forecasts, the diversity in earthquake magnitudes, and the differences between human and geological timelines all add to the challenges in reliable earthquake prediction (Mignan and Broccardo, 2020; Sun et al., 2022).
  While progress is being made, the emphasis in earthquake research has turned toward establishing effective earthquake forecast

models and early warning systems, understanding seismic risks, and improving preparation to lessen the effects of these deadly
 occurrences (Bose et al., 2008; Tiampo and Shcherbakov, 2012; Mousavi and Beroza, 2018; Mousavi et al., 2020; Tan et al.,
 2022). Nevertheless, with the advancements in field instrumentation, once an event occurs, we have attained the knowledge to
 estimate and record its information like magnitude, location, extent of ground shaking, etc., immediately (USGS (2023), IMD (2023)). The robustness of this data has also improved significantly over the years. An intriguing question here shall be, is it possible to predict and be better prepared for a forthcoming event using these information? This study tackles this problem by

45 compiling an extensive seismic dataset and building predictive models using state-of-the-art machine learning (ML) algorithms.

ML has evolved so much that its potential is widely explored to address numerous real-world problems (Schmidt et al., 2019; Kaushik et al., 2020; Sarker, 2021; Bertolini et al., 2021; Kumar et al., 2023b). Appropriate data processing using advanced ML algorithms led to successful prediction models. However, ML algorithms have only recently gained popularity in engineering seismology (Xie et al., 2020; Mousavi and Beroza, 2023). The most comprehensive application is in developing efficient Ground Motion Prediction Equations (GMPEs) (Alavi and Gandomi, 2011; Derras et al., 2014; Dhanya and Raghukanth, 2018; Gade et al., 2021; Seo et al., 2022; Sreenath et al., 2024). Moreover, several machine learning techniques have been explored, among those, Multilayer Perceptrons (MLPs) is the most widely used model in earthquake engineering applications (Xie et al., 2020). Raghukanth et al. (2017) utilised a similar model for suitably combining stationary and non-stationary parts of energy

55 series to forecast seismic energy. MLP technique is also widely used in developing ground motion prediction equations, as evidenced by the works of Derras et al. (2014); Dhanya and Raghukanth (2018, 2020); Douglas (2021). In another direction, Paolucci et al. (2018) proposed a simple MLP model that should efficiently generate broadband ground motions. Sharma et al. (2023a) improved the model by incorporating source, path and site characteristics. Another architecture, such as Linear Regression (LR), has also been applied in various seismological studies due to its simplicity and efficiency. Pairojn and Wasinrat

- 60 (2015) used LR for ground motion prediction in Thailand, while Cho et al. (2022) compared Artificial Neural Networks (ANN) and LR for predicting earthquake-induced slope displacement. The Random Forest (RF) technique has similarly motivated researchers across different fields, including seismology. Apart from the plain linear regression, studies have also used ridge regression for earthquake forecast problems Ahmed et al. (2024). Pyakurel et al. (2023) utilised five supervised algorithms, including RF, to predict earthquake-induced landslides for the 2015 Gorkha earthquake. Additionally, (Li and Goda, 2023)
- 65 extended the application of RF to tsunami early warning systems and loss forecasting. Furthermore, Support Vector Machines (SVM) with the optimised version named as Sequential Minimal Optimisation for regression (SMOreg), as proposed by Shevade et al. (2000), are widely used for parameter learning. This approach has been applied to various natural hazard contexts, such as flood susceptibility mapping (Saha et al., 2021), ground motion prediction equations (Altay et al., 2023), and landslide monitoring (Kumar et al., 2023b). Similar to SMOreg, instance-based learning is also well-explored in earthquake prediction
- 70 problems, as its reliability and accuracy are owed to the algorithm's resistance to noise and outliers, as well as its versatility in the use of distance measures. Its applicability in seismic prediction is well demonstrated by Reyes et al. (2013), Ghaedi and Ibrahim (2017), Al Banna et al. (2020), and Ridzwan and Yusoff (2023).

Apart from the individual machine learning techniques, ensemble learning is a mature and widely adopted methodology in the ML literature, which is renowned for averaging several models to enhance prediction accuracy and generality (Dietterich, 2000; De Gooijer and Hyndman, 2006; Alpaydin, 2007). Ensemble models combine different base learners based on techniques such as bagging, boosting, and stacking, and so maximize the variance in data, reduce overfitting, and improve model reliability. They have been successful across a variety of applications including medical diagnosis and climate modelling to financial forecasting (Re and Valentini, 2012; Tan et al., 2022; Rezaei et al., 2022). Although extensively used, their use in

seismic energy forecasting is still not well exploited, making the current research a timely and new addition in the geophysical hazard field.

The research is based on the success of ensemble learning with the application of a stacked ensemble framework that is specific to the challenge of seismic energy forecasting. Even though ensemble models are extensively applied in other areas such as health, climate, and finance, their niche use in seismic energy prediction remains limited and largely untapped. The predictions

- 85 of five individual machine learning models—MLP, RF, LR, SMOreg, and IBk—are combined and stacked in this research through a Random Forest meta-learner. In this setup, the final ensemble model uses the Random Forest as a meta-learner that synthesizes predictions from these individual models, each trained using domain-informed design choices and preprocessing. As such, the RF model mimics a consensus-based expert system, combining diverse perspectives across learning paradigms to enhance forecasting robustness.
- 90 The improved long-term seismic energy prediction capability—essential for forward-looking hazard mitigation—is the main contribution of this work. The proposed model showed versatility across diverse tectonic environments by working well for both global and western Himalayan data sets. The objective of this project is to apply stacked ensemble learning to develop

a reliable model for annual seismic energy predictions. The Empirical Mode Decomposition (EMD) method is applied to decompose global seismic energy time series, considering stationary and non-stationary components as inputs. The model is compared to existing research, and its predictive ability is established via a case study for the Western Himalaya region.

**2 Global Seismic Energy (GSE) time series**

95

Making accurate earthquake predictions requires a thorough earthquake catalog. We have used two global earthquake catalogs for this study. Raghukanth et al. (2017) used the ISC-GEM catalog (http://www.isc.ac.uk/) as the primary resource. In the current study, the model construction, comparison, and validation of the suggested approach are done using this catalog. A more thorough and current worldwide earthquake catalog (up to 2023) was created using the USGS seismic database (https://earthquake.usgs.gov/earthquakes/search/) and the ISC-GEM catalog after the methodology was verified using this data. For our analysis, we used the same inputs as those mentioned in Raghukanth et al. (2017). We provide a brief explanation of the processing that goes into creating the final time series that is used for modeling in order to improve understanding of the data. The validation catalog, which is the worldwide earthquake catalog that was obtained from Raghukanth et al. (2017), includes

- 105 data from 1900 to 2015, totaling 24,375 occurrences with a minimum magnitude of 4.98 Mw. The new global catalog created for this study, which will be referred to as the Global catalog, is compiled from both USGS and ISC-GEM sources and contains data from 1900 to 2023. Duplicates were thoroughly examined and eliminated because the data was obtained from two distinct sources. There were 988,812 distinct occurrences with a minimum magnitude of 1.09 after duplicate events were removed. All reported event magnitudes were converted to moment magnitude ( $M_w$ ) using empirical relationships from Scordilis (2006) and
- 110 Yenier et al. (2008) to guarantee magnitude uniformity across datasets. We next determined the magnitude of completeness  $(M_c)$  for both catalogs, which is the lowest magnitude above which earthquakes are consistently documented. According to Raghukanth et al. (2017), using the maximum curvature approach (Wiemer and Wyss, 2000),  $M_c$  was determined to be 6.4 for the validation catalog (Figure S1(a) in the supplemental document). Using the MATLAB-based program ZMAP version 7.1,  $M_c$  was calculated to be 4.9  $M_w$  for the Global catalog, as seen in Figure 1(a).
- 115 Furthermore, the Global catalog's year of completeness was established using the Stepp (1973) approach, showing that the catalog is complete starting in 1953 (Figure 1(b)). Figures S1(b) and S1(c) display the distribution of events from the full validation catalog (4,619 events), while Figure 1(c) shows the full Global catalog with 217,751 events. Because partial representation increases statistical bias, events with magnitudes  $